



# Preparing for an extensive $\Delta^{14}CO_2$ flask sample monitoring campaign over Europe to constrain fossil $CO_2$ emissions

Carlos Gómez-Ortiz[1], Guillaume Monteil[1,2], Ute Karstens[3], and Marko Scholze[1]

[1]Department of Physical Geography and Ecosystem Science, Lund University, Lund, Sweden.
[2]Barcelona Supercomputing Centre (BSC), Barcelona, Spain.
[3]ICOS Carbon Portal, Lund University, Lund, Sweden.

**Correspondence:** Carlos Gómez-Ortiz (carlos.gomez@nateko.lu.se)

**Abstract.** During 2024, an intensive $\Delta^{14}CO_2$ flask sampling campaign is being conducted at 12 sampling stations across Europe as part of the CO2MVS Research on Supplementary Observations (CORSO) project. These $\Delta^{14}CO_2$ samples, combined with $CO_2$ atmospheric measurements, are intended to enhance the estimation of fossil $CO_2$ emissions over Europe through inverse modeling. In this study, we perform a series of Observing System Simulation Experiments (OSSEs) to evaluate the

added value of such an intensive campaign as well as the different sampling strategies in estimating fossil fuel emissions. These strategies focus on selecting samples for inversions based on their fossil $CO_2$ and nuclear $^{14}C$ composition.

We explore three main sampling strategies: (1) a base case scenario using a uniform sampling approach without specific selection criteria, comparing current sampling methods with the inclusion of flask samples; (2) a strategy that selects samples with high fossil $CO_2$ contribution; and (3) a combined approach that also considers nuclear $^{14}C$ contamination to reduce

potential biases from nuclear facilities. In the first strategy, the results suggest that higher sampling density can improve the estimation of fossil $CO_2$ emissions, particularly during periods of low fossil fuel activity, such as in summer. This increase in sample quantity contributes to a reduction in uncertainty, enhancing the robustness of inverse modeling results. Furthermore, applying the strategy of selecting samples with high fossil $CO_2$ contamination shows potential for improving the accuracy of emission estimates. However, the most significant reduction in uncertainty is observed when the sampling strategy also

accounts for nuclear $^{14}C$ contamination. By considering nuclear emissions, this combined strategy helps to minimize potential biases, particularly in regions with high nuclear activity, such as France and the UK. The findings underscore the importance of not only increasing sample frequency but also carefully selecting samples based on their fossil and nuclear $CO_2$ composition to improve the reliability of fossil fuel emission estimates across Europe.

## 1 Introduction

On the path to refining our understanding of carbon dynamics and the anthropogenic contributions to atmospheric $CO_2$ levels, the technique of inverse modeling has emerged as a crucial tool. By integrating atmospheric observations of $CO_2$ with specific tracers measured *in situ* (e.g., $\Delta^{14}CO_2$, CO, APO) (Basu et al., 2020; Wang et al., 2020; Chawner et al., 2024) or remotely (e.g. $XCO_2$) (Fischer et al., 2017; Chen et al., 2023) inverse modeling enhances the distinction between fossil fuel emissions and natural biogeochemical fluxes. A leading example of such a tracer is radiocarbon ($^{14}C$) found in atmospheric $CO_2$, which



serves as a quantitative tracer to distinguish the fossil $CO_2$ from the biogenic component of the recently emitted $CO_2$ because fossil $CO_2$ is void of radiocarbon due to its half-life of 5,730 years, producing a reduction of the radiocarbon content of carbon ($\Delta^{14}CO_2$) (Levin et al., 2003). However, in Europe and other industrialized regions of the world, biogenic and fossil $CO_2$ signals can be of the same order of magnitude, and because additional signal masking can occur from pure $^{14}C$ emissions from nuclear processes (Graven, 2015), it is necessary to have precise $\Delta^{14}CO_2$ measurements (Levin et al., 2020). Graven and

Gruber (2011) found that in regions with a high influence of nuclear emissions such as Europe, North America, and East Asia, radiocarbon from these sources can offset around 20% of the fossil $CO_2$ dilution in $^{14}C$, which can translate into a potential bias in $CO_2$ attributed to fossil emissions (ffCO_2) larger than the bias caused by exchanges with the terrestrial biosphere over some areas. Vogel et al. (2013) in a local application in Toronto, Canada, found that this offset can be as high as 82% of the total annual fossil $CO_2$ emissions.

Data on nuclear facility emissions are generally limited to annual emissions, accessible through databases such as the European Commission RAdioactive Discharges Database (RADD) (https://europa.eu/radd/index.dox) or derived from energy production data from the Power Reactor Information System (PRIS) (https://pris.iaea.org/PRIS/home.aspx). These data sets often lack the high temporal resolution necessary to identify the possible effect of large emission events in radiocarbon samples. Studies such as those by Graven and Gruber (2011) and Zazzeri et al. (2018) provide essential emission factors and data, but

also highlight the high-resolution data availability gap we just mentioned. Strict data protection policies and security measures further compound the challenge of obtaining high-resolution time series data from nuclear facilities. Few studies have directly measured and reported emissions from nuclear facilities (Akata et al., 2013; Varga et al., 2020; Lehmuskoski et al., 2021) at higher temporal resolutions, such as daily or weekly. Vogel et al. (2013) for instance, found significant deviations in interannual timescales of nuclear emissions compared to emission factors reported by Graven and Gruber (2011), but a better agreement

with the long-term average observed for reactors in their study area. Most research examining the impact of nuclear emissions on ffCO_2 estimation is conducted in the vicinity of nuclear facilities, which allows sampling of winds directly coming from these facilities, reducing the need for high-resolution emission time series (Vogel et al., 2013; Kuderer et al., 2018). Consequently, the broader implications of nuclear emissions and their temporal variations on regional and continental scales remain less explored and understood, as evidenced in the study by Vogel et al. (2013). This localized focus limits our understanding of

the impact of nuclear facility emissions on ffCO_2 estimations on a continental scale, such as for Europe. In addition, in inverse modeling approaches that include both $CO_2$ and $\Delta^{14}CO_2$, the emissions from nuclear facilities are not optimized, leading to potential inaccuracies. The research by Bozhinova et al. (2014); Graven and Gruber (2011); Turnbull et al. (2011); Zazzeri et al. (2018) demonstrates this gap, suggesting the need for more sophisticated modeling and sampling approaches to integrate nuclear emissions accurately into atmospheric inversion techniques. In a sensitivity study by Maier et al. (2023), they found

that nuclear emissions could lead to a 25% low-biased ffCO_2 estimate if not corrected for, further emphasizing the need for accurate modeling and measurements.

In Europe, the Integrated Carbon Observation System (ICOS) Atmosphere network continuously measures $CO_2$ together with other greenhouse gases (GHG) at 38 stations in Europe. Additional tracers, as well as isotopes such as $^{13}C$ and $^{14}C$, are measured in periodic flask samples at 17 of these ICOS stations (see Figure 1). Most of the stations are located in remote





locations, where measurements are taken from tall towers of at least $100\mathrm{m}$ above ground level, on mountain tops, and on coastal sites (in the last two, measurements are usually taken a few meters above ground level). The objective of the network stations is that the measurements represent large areas, capturing signals of sources and sinks occurring even hundreds of kilometers from the station. Currently, $^{14}C$ is measured mainly in two-weekly integrated flask samples, at the highest sampling height available at each station (red and yellow dots in Figure 1). Since 2016, some stations such as Hyltemossa (HTM) in Sweden and Gartow (GAT) in Germany have been taking 1 hour $^{14}C$ flask samples every third day at 13:00 local time. Of the approximately 100 flask samples that are taken at these stations during the year as quality control for continuous measurements and for the analysis of other tracers and isotopes, around 25 are selected to analyze $\Delta^{14}CO_2$ to be used for the estimation of ffCO$_2$. Levin et al. (2020) designed a strategy to choose samples that mainly captured large events of fossil fuel CO$_2$ contamination for their posterior analysis of $\Delta^{14}CO_2$. They suggested defining a threshold for the mixing ratio of the fossil fuel component CO$_2$ (ffCO$_2$) and for the enhancement of CO (CO is a co-emmitted species from fossil fuel burning) relative to the background mixing ratio at the time the flask sample is taken. This can be determined by near-real-time (NRT) atmospheric transport simulations (for ffCO$_2$ and ffCO) or by using continuous observations of CO at the station. As part of the CO2MVS Research on Supplementary Observations (CORSO) project (https://www.corso-project.eu/) funded by the Horizon Europe program of the European Commission, an intensive sampling campaign of $\Delta^{14}CO_2$ is carried out in 2024. In this project, flask samples are taken approximately every three days, completely dedicated to the analysis of $\Delta^{14}CO_2$ at 10 of the current ICOS sampling stations around Sweden, Germany, the Netherlands, France and the Czech Republic, complemented by two additional stations in Poland (Białystok) and England (Heathfield), and three background stations that take 2-weekly integrated samples in Ireland (Mace Head), Spain (Izaña), and Canada (Alert).

In this paper, we investigate the impact of combining intensive sampling with regular integrated sampling for estimating fossil CO$_2$ emissions on a subregional and subannual scale. We use the multi-tracer enabled version of the Lund University Modular Inversion Algorithm (LUMIA) system (Gómez-Ortiz et al., 2023) by performing a series of perfect transport Observing System Simulation Experiments (OSSEs). The study aims to address three key research questions: 1) What is the added value of intensive $\Delta^{14}CO_2$ sampling compared to the current sampling done in ICOS? 2) Is there a benefit in selecting $\Delta^{14}CO_2$ flask samples based on their fossil contribution to improve fossil CO$_2$ emissions estimates? 3) Does further selection of flask samples based on nuclear contamination provide additional benefits when estimating fossil CO$_2$ emissions?

To address these questions, we calculate a series of synthetic observations by performing a forward simulation of the transport model with a set of assumed true fluxes. We then select the observations based on various sampling strategies, including uniform, fossil CO$_2$-based, and nuclear contamination-based selection criteria. Subsequently, these observations are inverted using LUMIA to estimate fossil CO$_2$ emissions, allowing us to quantify the differences in bias and uncertainty of the different sampling strategies. This approach enables a comprehensive evaluation of how intensive sampling and targeted sample selection can enhance the estimation of fossil CO$_2$ emissions at both subregional and subannual scales, ultimately providing insights into optimizing future sampling strategies for more accurate greenhouse gas monitoring.





**Figure 1.** Sampling stations selected for this study and their identification according to the measured tracers and their participation in the CORSO project (dark blue diamonds). Green dots represent the stations where only $CO_2$ is measured, yellow dots where additionally $\Delta^{14}CO_2$ is measured in 1-hour flasks and red dots where $\Delta^{14}CO_2$ is measured in approximately 2-weekly integrated samples.



**Table 1.** Sampling sites include in this study and $\Delta^{14}CO_2$ sampling type according to the current status and the CORSO project.

| Site | Name | Country | Latitude | Longitude | Altitude | Sampling height | CORSO | Current $\Delta^{14}CO_2$ sampling | CORSO $\Delta^{14}CO_2$ sampling |
|---|---|---|---|---|---|---|---|---|---|
| BIK | Białystok | PL | 53.2294 | 23.0128 | 183.0 | 300.0 | X | | Flask |
| BIR | Birkenes | NO | 58.3886 | 8.2519 | 219.0 | 75.0 | | | |
| CBW | Cabauw | NL | 51.9703 | 4.9264 | 0.0 | 207.0 | X | Integrated | Flask |
| CMN | Monte Cimone | IT | 44.1936 | 10.6999 | 2165.0 | 8.0 | | | |
| GAT | Gartow | DE | 53.0657 | 11.4429 | 70.0 | 341.0 | X | Integrated | Flask |
| HEL | Helgoland | DE | 54.1804 | 7.8833 | 43.0 | 110.0 | | | |
| HFD | Heathfield | GB | 50.9770 | 0.2310 | 157.3 | 100.0 | X | | Flask |
| HPB | Hohenpeissenberg | DE | 47.8011 | 11.0246 | 934.0 | 131.0 | X | Integrated | Flask |
| HTM | Hyltemossa | SE | 56.0976 | 13.4189 | 115.0 | 150.0 | X | Integrated | Flask |
| IPR | Ispra | IT | 45.8147 | 8.6360 | 210.0 | 100.0 | | | |
| JFJ | Jungfraujoch | CH | 46.5475 | 7.9851 | 3571.8 | 13.9 | | Integrated | |
| JUE | Jülich | DE | 50.9102 | 6.4096 | 98.0 | 120.0 | | | |
| KIT | Karlsruhe | DE | 49.0915 | 8.4249 | 110.0 | 200.0 | X | Integrated | Flask |
| KRE | Křešín u Pacova | CZ | 49.5720 | 15.0800 | 534.0 | 250.0 | X | Integrated | Flask |
| LIN | Lindenberg | DE | 52.1663 | 14.1226 | 73.0 | 98.0 | X | Integrated | Flask |
| LMP | Lampedusa | IT | 35.5181 | 12.6322 | 45.0 | 8.0 | | | |
| LUT | Lutjewad | NL | 53.4036 | 6.3528 | 1.0 | 60.0 | | | |
| MHD | Mace Head | IE | 53.3261 | -9.9036 | 5.0 | 24.0 | X | | Integrated |
| NOR | Norunda | SE | 60.0864 | 17.4794 | 46.0 | 100.0 | | Integrated | |
| OPE | Observatoire pérenne de l'environnement | FR | 48.5619 | 5.5036 | 390.0 | 120.0 | X | Integrated | Flask |
| OXK | Ochsenkopf | DE | 50.0300 | 11.8083 | 1022.0 | 163.0 | | Integrated | |
| PAL | Pallas | FI | 67.9733 | 24.1157 | 565.0 | 12.0 | | Integrated | |
| PRS | Plateau Rosa | IT | 45.9300 | 7.7000 | 3480.0 | 10.0 | | | |
| PUI | Puijo | FI | 62.9096 | 27.6549 | 232.0 | 84.0 | | | |
| PUY | Puy de Dôme | FR | 45.7719 | 2.9658 | 1465.0 | 10.0 | | | |
| SAC | Saclay | FR | 48.7227 | 2.1420 | 160.0 | 100.0 | | Integrated | |
| SMR | Hyytiälä | FI | 61.8474 | 24.2947 | 181.0 | 125.0 | | | |
| STE | Steinkimmen | DE | 53.0431 | 8.4588 | 29.0 | 252.0 | X | Integrated | Flask |
| SVB | Svartberget | SE | 64.2560 | 19.7750 | 269.0 | 150.0 | | Integrated | |
| TOH | Torfhaus | DE | 51.8088 | 10.5350 | 801.0 | 147.0 | | | |
| TRN | Trainou | FR | 47.9647 | 2.1125 | 131.0 | 180.0 | X | Integrated | Flask |
| UTO | Utö - Baltic sea | FI | 59.7839 | 21.3672 | 8.0 | 57.0 | | | |
| WAO | Weybourne | GB | 52.9500 | 1.1210 | 31.0 | 10.0 | | | |
| WES | Westerland | DE | 54.9231 | 8.3080 | 12.0 | 14.0 | | | |
| ZSF | Zugspitze | DE | 47.4165 | 10.9796 | 2666.0 | 3.0 | | | |



## 2 The LUMIA framework

We use the Lund University Modular Inverse Algorithm (LUMIA) (Monteil and Scholze, 2021) to perform $CO_2$ and $\Delta^{14}CO_2$

perfect-transport Observing System Simulation Experiments (OSSEs) for the year 2018 covering Europe in a regional domain ranging from 15°W, 33°N to 35°E, 73°N, as shown in Figure 1, similar to previous regional European inverse modeling studies(Monteil et al., 2020; Thompson et al., 2020). LUMIA is an inversion framework originally designed for regional $CO_2$ inversions in Europe. The framework was later extended to perform simultaneous inversions of $CO_2$ and $\Delta^{14}CO_2$ to estimate fossil $CO_2$ emissions over Europe (Gómez-Ortiz et al., 2023), which we use in this study with minor modifications detailed in

this section. Since the initial release of LUMIA, it has incorporated the two-step atmospheric inversion scheme proposed by Rödenbeck et al. (2009), as thoroughly explained by Monteil and Scholze (2021). In this approach, for each observation (either $CO_2$ or $\Delta^{14}CO_2$), the modeled mixing ratio $y^{\mathrm{m}}$ is described as the total of the contributions of the "foreground" $y^{\mathrm{f}}$ (mixing ratios due to fluxes directly related with $y^{\mathrm{m}}$ by the model, limited spatially by the domain and temporally by the length of the simulation) and the "background" $y^{\mathrm{b}}$ (i.e., any additional contribution not captured by the foreground fluxes, including external

sources or preexisting atmospheric mixing ratios):

$$y^{\mathrm{m}} = y^{\mathrm{b}} + y^{\mathrm{f}} \tag{1}$$

which can be expanded for each tracer ($CO_2$ and $\Delta^{14}CO_2$) as:

$$y^{\mathrm{m}}_{CO_2} = y^{\mathrm{b}}_{CO_2} + y^{\mathrm{f}}_{\mathrm{ff}} + y^{\mathrm{f}}_{\mathrm{bio}} + y^{\mathrm{f}}_{\mathrm{oce}} \tag{2a}$$

$$y^{\mathrm{m}}_{C\Delta^{14}C} = \underbrace{y^{\mathrm{b}}_{C\Delta^{14}C} + y^{\mathrm{b}}_{\mathrm{cosmo}}}_{\text{background}} + \underbrace{y^{\mathrm{f}}_{\Delta\mathrm{ff}} + y^{\mathrm{f}}_{\Delta\mathrm{bio}} + y^{\mathrm{f}}_{\Delta\mathrm{oce}} + y^{\mathrm{f}}_{\mathrm{biodis}} + y^{\mathrm{f}}_{\mathrm{ocedis}} + y^{\mathrm{f}}_{\mathrm{nuc}}}_{\text{foreground}} \tag{2b}$$

where $y^{\mathrm{m}}_{CO_2}$ is the modeled $CO_2$ mixing ratio and $y^{\mathrm{b}}_{CO_2}$ is the background $CO_2$ mixing ratio. On the right-hand side of Eq. 2a, $y^{\mathrm{f}}_{\mathrm{ff}}$ is the mixing ratio within the domain due to fossil $CO_2$ ($F_{\mathrm{ff}}$), $y^{\mathrm{f}}_{\mathrm{bio}}$ the mixing ratio due to the net exchange of $CO_2$ between the atmosphere and terrestrial ecosystems (Net Ecosystem Exchange, NEE, hereafter biosphere flux, $F_{\mathrm{bio}}$) , and $y^{\mathrm{f}}_{\mathrm{oce}}$ the mixing ratio due to the net exchange of $CO_2$ between the atmosphere and oceans ($F_{\mathrm{oce}}$).

All terms in Eq. 2b are in units of $CO_2 \times \Delta^{14}CO_2$ (e.g. ppm‰) or $C\Delta^{14}C$ for simplification, since the values in ‰ are not additive (see Basu et al. (2016) and Gómez-Ortiz et al. (2023) for additional details). In this equation, $y^{\mathrm{m}}_{C\Delta^{14}C}$ and $y^{\mathrm{b}}_{C\Delta^{14}C}$ are the modeled and background $C\Delta^{14}C$ mixing ratios, respectively. $y^{\mathrm{b}}_{\mathrm{cosmo}}$ is the $C\Delta^{14}C$ mixing ratio due to the cosmogenic production of $^{14}CO_2$ in the stratosphere ($F_{\mathrm{cosmo}}$). $y^{\mathrm{b}}_{\mathrm{cosmo}}$ is accounted in the background ($y^{\mathrm{b}}_{C\Delta^{14}C}$), since LUMIA was designed to assimilate only surface fluxes. Furthermore, on a regional scale, sampling sites are considered to be similarly influenced by

$^{14}C$-enriched stratospheric air and its influence on tropospheric $^{14}C$ can be neglected (Maier et al., 2023; Lingenfelter, 1963).





A large influence of $^{14}$C cosmogenic production can be expected in stations sampling close to the low stratosphere (above 6km) (Turnbull et al., 2009) which is not the case for any of the stations considered in this study (see Fig. 1).

The first foreground term in Eq. 2b, $\boldsymbol{y}^{\mathrm{f}}_{\Delta\mathrm{ff}}$, represents the C$\Delta^{14}$C mixing ratio (or dilution of it) due to the absence of $^{14}$C in fossil $CO_2$. Fossil $CO_2$ is devoid of $^{14}$C that has decayed after being buried for millions of years ($t^{^{14}\mathrm{C}}_{\frac{1}{2}} \approx 5730$ years), but
has an impact on the atmosphere $\Delta^{14}CO_2$, diluting the existing $^{14}CO_2$ into more $^{12}CO_2$. This dilution effect is modeled by transporting a tracer $\boldsymbol{y}^{\mathrm{f}}_{\Delta\mathrm{ff}}$ with a value of $\Delta^{14}$C of $-1000‰$ (which corresponds to a $^{14}$C/$^{12}$C ratio of 0).

The next terms, $\boldsymbol{y}^{\mathrm{f}}_{\Delta\mathrm{bio}}$ and $\boldsymbol{y}^{\mathrm{f}}_{\Delta\mathrm{oce}}$ represent the net exchange from the atmosphere with the biosphere and the ocean, respectively. The contribution of these exchanges is modeled by transporting the biosphere and ocean fluxes multiplied by the isotope signature of the current atmosphere. $\boldsymbol{y}^{\mathrm{f}}_{\mathrm{biodis}}$ and $\boldsymbol{y}^{\mathrm{f}}_{\mathrm{ocedis}}$ are the contributions due to isotopic disequilibrium. The old carbon
that has been stored for many years in the biosphere and the ocean has a different isotopic signature compared to the current atmosphere. When this carbon is released from the source to the atmosphere (for the biosphere mainly due to heterotrophic respiration), it creates disturbances in the atmospheric isotopic composition. The carbon released from the biosphere is mainly $^{14}$C enriched carbon captured after the atmospheric nuclear bomb tests of the 1960s, while the ocean releases mainly $^{14}$C depleted carbon that has been at the bottom of the ocean long enough to decay to signatures lower than the current atmosphere.

The last term, $\boldsymbol{y}^{\mathrm{f}}_{\mathrm{nuc}}$, represents the contribution due to the radiocarbon emissions generated by nuclear activities ($\boldsymbol{F}_{\mathrm{nuc}}$), mainly from nuclear facilities such as nuclear power plants and spent fuel reprocessing plants, since the contribution of nuclear bomb tests is now considered depleted (Levin et al., 2020).

As in the original implementation of LUMIA, here we use the global TM5 model (Huijnen et al., 2010) to calculate the background mixing ratios ($\boldsymbol{y}^{\mathrm{b}}$) and the Lagrangian FLEXPART model (Pisso et al., 2019) to perform the regional transport
($\boldsymbol{y}^{\mathrm{f}}$) and the inversions. In the following sections, we explain further the implementation of the models.

## 2.1  Background mixing ratios (TM5)

Background mixing ratios are the portion of $CO_2$ or $\Delta^{14}CO_2$ in the atmosphere that originates from sources outside the study domain. This can be a combination of emissions transported by large-scale atmospheric circulation, regional transport from outside the domain, and air masses reentering the domain (Rödenbeck et al., 2009). In this study, we use the implementation
of the background mixing ratio calculation in TM5-4DVar developed by Monteil and Scholze (2021) based on the methodology proposed by Rödenbeck et al. (2009), integrated with the implementation of TM5-4DVar to include $CO_2$ or $\Delta^{14}CO_2$ developed by Basu et al. (2016) (https://sourceforge.net/p/tm5/cy3_4dvar/ci/default/tree/proj/tracer/radio_co2/, last visited in August 2024). Here, we model the background mixing ratio using global optimized fluxes and an initial condition from Basu et al. (2020) for 2010. These fluxes are in a horizontal resolution of $3° \times 2°$ (25 hybrid sigma-pressure vertical levels for $\boldsymbol{F}_{\mathrm{cosmo}}$),
and variable time resolutions for the individual fluxes: 1 hour for $\boldsymbol{F}_{\mathrm{ff}}$, 3 hours for $\boldsymbol{F}_{\mathrm{bio}}$ and $\boldsymbol{F}_{\mathrm{oce}}$, 1 month for $\boldsymbol{F}_{\mathrm{biodis}}$ and $\boldsymbol{F}_{\mathrm{ocedis}}$, and 1 year for $\boldsymbol{F}_{\mathrm{nuc}}$ and $\boldsymbol{F}_{\mathrm{cosmo}}$. The simulation is driven by meteorological fields from the European Centre for Medium-Range Weather Forecasts (ECMWF) ERA5 reanalysis project (Hersbach et al., 2020).

Here, we describe a small modification to the original implementation by Monteil and Scholze (2021) to account for the cosmogenic production in $\boldsymbol{y}^{\mathrm{b}}_{\mathrm{C}\Delta^{14}\mathrm{C}}$ (see Sec. 2, Eq. 2b):



The background components $\boldsymbol{y}^{\mathrm{b}}_{\mathrm{CO_2}}$ and $\boldsymbol{y}^{\mathrm{b}}_{\mathrm{C\Delta^{14}C}}$ are calculated as follows:

1. A global forward run with TM5 to calculate the mixing ratio fields $\boldsymbol{y}^{\mathrm{TM5}}_{\mathrm{CO_2}}$ and $\boldsymbol{y}^{\mathrm{TM5}}_{\mathrm{C\Delta^{14}C}}$.

2. A modified TM5 forward run where all fluxes and mixing ratios are set to zero in all time steps outside the regional domain (Fig. 1) to calculate $\boldsymbol{y}^{\mathrm{f,\,TM5}}_{\mathrm{CO_2}}$.

3. For calculating $\boldsymbol{y}^{\mathrm{f,\,TM5}}_{\mathrm{C\Delta^{14}C}}$, in addition to Step 2, $\boldsymbol{F}_{\mathrm{cosmo}}$ is set globally to zero in order to keep it in the background in the next step.

4. The background mixing ratios are calculated as: $\boldsymbol{y}^{\mathrm{b}}_{\mathrm{t}} = \boldsymbol{y}^{\mathrm{TM5}}_{\mathrm{t}} - \boldsymbol{y}^{\mathrm{f,\,TM5}}_{\mathrm{t}}$, with t indicating the tracers $CO_2$ and $C\Delta^{14}C$.

## 2.2 Regional transport (FLEXPART)

Following the methodology described in Monteil and Scholze (2021) and Gómez-Ortiz et al. (2023), our regional transport model (i.e. the operator to calculate $\boldsymbol{y}^{\mathrm{f}}$ in Equations 1 and 2) is composed of a series of pre-computed footprints with FLEX-PART (Pisso et al., 2019) driven by ERA5 reanalysis data for 2018 at a spatio-temporal resolution of $0.25° \times 0.25°$ and 1 hour, using the Python code developed to run and post-process the footprints to be used in LUMIA (https://github.com/lumia-dev/runflex, last accessed in August 2024). We compute two types of footprints: instant or flask footprints to simulate continuous $CO_2$ and CO observations (the latter used only for sampling selection as described in Sec. 3.5), and flask $\Delta^{14}CO_2$ samples, and integrated footprints to simulate $\Delta^{14}CO_2$ integrated observations (see Sec.

We compute instant footprints from the observation time and 14 days back in time releasing 10000 particles (Monteil and Scholze, 2021), and we use the same footprint to model $CO_2$, CO, and $\Delta^{14}CO_2$ at the corresponding observation time and sampling station. These footprints are computed for a passive air tracer, i.e. without any atmospheric chemistry reactions. Therefore, for CO we only evaluate the regional contributions (Levin et al., 2020) without accounting for the background and reactions with other atmospheric components. For the integrated footprints, we set a fixed integration time of 2 weeks (14 days), distribute 10000 FLEXPART particles over this integration period, and then simulate 14 days backward from the integration start time (Gómez-Ortiz et al., 2023).

## 2.3 The inverse modeling problem

LUMIA follows an implementation of the variational approach (4D-Var). This approach seeks to iteratively minimize the mismatch between the model output and observations $\delta_{\boldsymbol{y}}$ by optimizing the control vector $\boldsymbol{x}$. The optimization process is guided by a cost function, $J(\boldsymbol{x})$, defined as:

$$J(\boldsymbol{x}) = \frac{1}{2}\left(\boldsymbol{x} - \boldsymbol{x}^{\mathrm{b}}\right)^{T}\mathbf{B}^{-1}\left(\boldsymbol{x} - \boldsymbol{x}^{\mathrm{b}}\right) + \frac{1}{2}\left(\mathbf{H}\boldsymbol{x} - \delta_{\boldsymbol{y}}\right)^{T}\mathbf{R}^{-1}\left(\mathbf{H}\boldsymbol{x} - \delta_{\boldsymbol{y}}\right) \tag{3}$$

In this equation, $\boldsymbol{x}^{\mathrm{b}}$ represents the prior estimate of the control vector, $\mathbf{B}$ is the prior uncertainty covariance matrix, $\mathbf{R}$ is the observational uncertainty covariance matrix, and $\mathbf{H}$ is the Jacobian of the observation operator $H$ which includes the transport





model itself (i.e., pre-computed footprints described in Sec. 2.2) and other steps needed to express $y$ as a function of $x$ (e.g.,
aggregation and disaggregation of flux components, accounting for the boundary conditions, etc.).

The control vector $x$ contains the set of parameters adjustable by the inversion, which are offsets to the different sources and sinks of $CO_2$ and $\Delta^{14}CO_2$ that we want to estimate (for this study, the fossil and biosphere $CO_2$ fluxes). We solve for clusters that are aggregated in time and space. These clusters are formed based on the sensitivity of the observation network to emissions from different regions. High-resolution optimization is applied to areas directly upwind of sampling stations, while
regions with lower sensitivity are optimized at a coarser resolution.

The prior error covariance matrix ($\mathbf{B}$) is constructed in three steps. First, the variances are determined to represent the assumed spatio-temporal uncertainties of the fluxes. Next, covariances are calculated based on assumed spatial and temporal correlations, incorporating the distance between grid clusters and the time difference between flux intervals. Finally, the entire matrix is scaled using a uniform factor to match category-specific annual uncertainty values. The formulas used for fossil $CO_2$
emissions differ from those used for other fluxes to account for better-known emission locations and to avoid artificially low uncertainties due to flux compensations (Gómez-Ortiz et al., 2023).

The observation uncertainty matrix ($\mathbf{R}$) includes both measurement uncertainties and model representation uncertainties, which account for the model's inability to perfectly simulate observations even with accurate fluxes. Ideally, the diagonal of $\mathbf{R}$ holds the total uncertainty for each observation, while the off-diagonals represent error correlations between observations.
However, since these correlations are hard to quantify, common practice is to set these error correlations (off-diagonal elements) to zero. The observation uncertainty can then be provided as a simplified observation error vector (Monteil and Scholze, 2021).

The iterative procedure works by adjusting $x$ to minimize the cost function $J(x)$. The optimal solution is achieved when the gradient of the cost function, $\nabla_x J$, is close to zero. This approach ensures that the final estimate of $x$ provides the best possible fit to the synthetic observational data while taking into account the uncertainties in both the prior information and the
observations (Gómez-Ortiz et al., 2023).

## 3 Experimental design

In this paper, we focus on the implementation of perfect transport Observing System Simulation Experiments (hereafter OSSEs). In OSSEs, we calculate a series of synthetic observations, using a set of assumed "true" fluxes ($F^{\mathrm{t}}$), by performing a forward run of our transport model. Afterwards, using a set of "prior" fluxes, we can evaluate how well the inversion
framework performs in recovering the assumed "true" fluxes. In this case, perfect transport means that we use the same transport model to produce the synthetic observations and to perform the atmospheric inversions, as well as the same background for the synthetic observations and the modeled mixing ratios. In this section, we describe the flux products used as true and prior fluxes (Sec. 3.1), the calculation of the synthetic observations (Sec. 3.3), the model setup (i.e., the information needed to construct the matrices $\mathbf{B}$ and $\mathbf{R}$ and the control vector $x$) (Sec. 3.4), the selection criteria of the synthetic $\Delta^{14}CO_2$ flask
samples (Sec. 3.5), and the design of the OSSEs (Sec. 3.6).



## 3.1 True, prior and prescribed fluxes

The assumed true fluxes, denoted as $F^t$, are used to generate synthetic observations through a forward run of our transport model. For the global transport simulation, we use the posterior fluxes from Basu et al. (2020), as explained in Sec. 2.1. For the regional transport, all fluxes have a resolution of $0.5° \times 0.5°$ and 1 hour in the domain shown in Figure 1.

We use as true fossil $CO_2$ flux ($F^t_{ff}$) a product (Koch and Gerbig, 2023) for 2018 based on the Emission Database for Global Atmospheric Research (EDGAR) version 4.3.2 emission product (Janssens-Maenhout et al., 2019) following temporal variations based on MACC-TNO Denier van der Gon et al. (2011) and with temporal extrapolations and disaggregation using the COFFEE approach (Steinbach et al., 2011). For the selection of the $\Delta^{14}CO_2$ flask samples, we use a fossil CO product based on the same methodology described for $F^t_{ff}$.

As true biosphere fluxes ($F^t_{bio}$), we use a simulation for 2018 of the LPJ-GUESS vegetation model (Wu, 2023; Smith et al., 2014) , the Jena Carbo-Scope oc_v2020 product based on the SOCAT data set of $pCO_2$ observations (van der Woude et al., 2022; Rödenbeck et al., 2022, 2013) as true ocean fluxes ($F^t_{oce}$), and as true terrestrial and oceanic isotopic disequilibrium fluxes ($F^t_{biodis}$ and $F^t_{ocedis}$) we use the optimized fluxes from Basu et al. (2020) regridded to match the spatial and temporal resolution of the regional transport. Both disequilibrium fluxes are prescribed in the experiments, and hence they are not

optimized. This decision is due to the high uncertainty derived from optimizing $F_{biodis}$, and the low impact of $F_{oce}$ and $F_{ocedis}$ in the study domain, as we found in a previous study (Gómez-Ortiz et al., 2023). The emission products from nuclear facilities are described in detail in the next section (Sec. 3.2).

     As prior fluxes, we use the Open-source Data Inventory for Anthropogenic $CO_2$ (ODIAC) (Oda et al., 2018) for 2018 (Oda and Maksyutov, 2020) to represent prior fossil $CO_2$ emissions ($F_{ff}$). For prior biosphere emissions ($F_{bio}$), we use fluxes

simulated by the Vegetation Photosynthesis and Respiration Model (VPRM) (Mahadevan et al., 2008; Thompson et al., 2020) for the year 2018 (Gerbig and Koch, 2021).

## 3.2 Radiocarbon emissions from nuclear facilities ($F_{nuc}$)

Nuclear $^{14}C$ fluxes ($F_{nuc}$) are generally prescribed in inverse modeling studies due to the high uncertainty derived from the lack of information on temporal variability. However, it has been shown in previous studies that nuclear emissions can have

a large impact on the estimation of fossil $CO_2$ emissions (Bozhinova et al., 2014; Graven and Gruber, 2011; Turnbull et al., 2011; Zazzeri et al., 2018). For this reason, we produced sets of nuclear fluxes: one with a temporal variability to be used as the true flux ($F^t_{nuc}$), and the second one with the emissions evenly distributed throughout the year as is usual for this flux category (Basu et al., 2016, 2020; Gómez-Ortiz et al., 2023). Both flux products are based on the data (Storm et al., 2024) used and described in Maier et al. (2023), therefore, both products have the same annual budget and spatial distribution, the latter using

the location of the nuclear facilities and aggregated over the $0.5° \times 0.5°$ grid.

     For the temporal distribution of $F^t_{nuc}$, we use the weekly temporal profiles reported by Varga et al. (2020) for the Paks Nuclear Power Plant (NPP) in Hungary and the monthly profiles reported by Akata et al. (2013) for the Rokkasho Spent Fuel Reprocessing Plant (SFR) in Japan. Both studies reported at least three years of temporal profiles. Therefore, we assign the





temporal profile by randomly selecting a time span corresponding to a year starting from a random date and then assigning
it to the corresponding type of nuclear facility (NPP or SFR). We did this because we did not find any evident seasonality in
the temporal profiles of these two studies, and, in addition, such temporal profiles can vary between different types of nuclear
reactors. With this temporal distribution, we want to add extra variability to the nuclear contribution to atmospheric $\Delta^{14}CO_2$
and study its impact when using the prescribed flat-year nuclear emissions to estimate fossil $CO_2$ emissions. However, we
are aware of the differences among the types of nuclear facilities and how this can affect the temporal profile. As mentioned
previously, for the prescribed flux, we incorporate a flat-year nuclear emission product. This approach allows the inversion to
follow a traditional approach, yet still introduces a representation of non-perfect nuclear emissions into the model.

### 3.3  Synthetic observations

The background component of the synthetic observations ($y^b$ in Eq. 2) is calculated as explained in Sec. 2.1 using the fluxes
described in Sec. 3.1. We calculate hourly mixing ratios for each sampling station. For the flask ($\Delta^{14}CO_2$) samples and the
instant ($CO_2$) observations, the background is the model output at each observation time. For the integrated $\Delta^{14}CO_2$ samples,
the background is calculated as the average of the mixing ratios computed from the start date of the sampling to the end date
of the integration period (14 days for this study).

With the instant and integrated footprints described in Sec. 2.2, we perform a forward run of our regional model using
the true fluxes introduced in Section 3.1 to generate mixing ratio time series of $CO_2$, CO, and $\Delta^{14}CO_2$. We use a CO flux
product based on the same methodology as the fossil $CO_2$ product (see Sec. 3.1) to simulate the CO mixing ratio and perform
the $\Delta^{14}CO_2$ sample selection following the methodology described in Levin et al. (2020). As a final step, we add a random
perturbation to the synthetic observations ( $CO_2$ and $\Delta^{14}CO_2$) without exceeding the assumed observation uncertainty to
mitigate the assumption of a perfect transport model.

As in previous studies (Monteil and Scholze, 2021; Gómez-Ortiz et al., 2023), we select the $CO_2$ synthetic observations
for the times of the day when we can get a good model representation, as usually done in real atmospheric inversions. This is
between 11:00 and 15:00 local time (LT), when the boundary layer is most likely well-developed, for sampling locations below
1000 m.a.s.l, and between 22:00 and 2:00 LT for mountaintop sampling stations, when the boundary layer is most likely below
the sampling intake and the free troposphere is sampled.

### 3.4  Model setup

275  We use the same model setup for all the OSSEs described in Sec. 3.6. As mentioned above, in all experiments, we optimize
only the fossil and biosphere $CO_2$ fluxes ($F_{ff}$ and $F_{bio}$, respectively). The control vector $x$ is composed of clusters of 2500 grid
points and weekly offsets for each flux category.

For the construction of the prior error covariance matrix B, we assume an exponential temporal correlation of one month
for both fluxes and a Gaussian spatial correlation of 200 km for $F_{ff}$ and 500 km for $F_{bio}$. We assume a prior uncertainty of
280  0.21 PgC yr$^{-1}$ (30% of the prior annual budget) for $F_{ff}$ and 0.37 PgC yr$^{-1}$ (25% of the absolute prior annual budget) for $F_{bio}$
aggregated over our European domain. For $F_{ff}$, this uncertainty reflects the difference in the 2018 annual budget for the study




domain between two commonly used fossil $CO_2$ emission products: EDGAR ($\boldsymbol{F}_{ff}^{t}$) and ODIAC. For $\boldsymbol{F}_{bio}$, the value represents the maximum difference between the simulations for 2018 of two ecosystem models, LPJ-GUESS and VPRM, observed in July due to the peak in biospheric production. However, since in a real inversion we do not have information regarding the spatial and temporal distribution of the emissions beyond the prior estimates, the distribution of the uncertainties is chosen arbitrarily and weighted by the magnitudes of the prior estimate. We aim for higher uncertainties where we have higher fluxes, but we also want the model to have the flexibility to adjust in areas with lower fluxes. In the case of $\boldsymbol{F}_{ff}$, the uncertainty is distributed by weighting the value in each grid cell by the ratio $\log(\text{Daily total})/\text{Daily total}$ of that grid cell. For $\boldsymbol{F}_{bio}$, the uncertainty is distributed proportionally to the square root of the absolute value of each grid cell and time step.

For the construction of the observational uncertainty covariance matrix $\mathbf{R}$, we calculate the observation errors as follows: for $CO_2$ observations, the prior error for each observation is set to the standard deviation of observations within a $\pm 3.5$ day window around it, while for $\Delta^{14}CO_2$ observations (both integrated and flask samples), we use a constant value of $0.9$ ppm $C\Delta^{14}C$ ($2.15 \pm 0.05$ ‰ $\Delta^{14}CO_2$).

## 3.5 Synthetic $\Delta^{14}CO_2$ flask sample selection

There are three key criteria for selecting $\Delta^{14}CO_2$ flask samples: 1) samples taken at midday (13:00 LT) approximately every third day, 2) samples that capture events of high fossil $CO_2$ contamination, and 3) samples that avoid events of high nuclear emissions. Sampling at midday ensures strong atmospheric mixing, reducing model transport errors and providing stable, low-variability conditions for accurate quality control. Capturing events of high fossil $CO_2$ emissions involves selecting samples based on thresholds for the mixing ratios of fossil $CO_2$ and fossil CO. Fossil CO is a reliable tracer of fossil $CO_2$ because it is co-emitted during combustion processes but is not influenced by biological activity (Levin et al., 2020). Avoiding high nuclear emissions is crucial to prevent masking the fossil fuel signal with nuclear $^{14}CO_2$ emissions (Maier et al., 2023; Graven and Gruber, 2011).

For the $\Delta^{14}CO_2$ flask sample selection, we follow the same thresholds for fossil $CO_2$ ($\geq 4$ ppm) and fossil CO ($\geq 40$ ppb) (hereafter ffCO2 and ffCO, respectively) as proposed by Levin et al. (2020) to capture events of high fossil $CO_2$ contamination. Additionally, we introduce a new threshold for nuclear $C\Delta^{14}C$ of $\leq 1$ ppm $C\Delta^{14}C$ to avoid capturing events of high nuclear $\Delta^{14}CO_2$ contamination. This value is based on forward runs using both nuclear emission products (with and without a temporal profile). At sites not directly influenced by nuclear emissions, such as Białystok (BIK, see Fig. 1 and Table 1), this threshold represents $87\%$ of the synthetic observations at 13:00 local time for the year 2018. In contrast, at sites with high nuclear impact, such as Karlsruhe (KIT) in Germany, it represents $41\%$ of the synthetic observations (see Fig. 2). This value is an approximation based on the synthetic data and forward simulations since we do not know the real behavior and magnitude of the nuclear emissions. Since LUMIA calculates the individual contribution of each flux category in Equation 2 to the mixing ratio of their corresponding air tracer, we use these values to implement the sampling selection strategies.

Approximately 120 $\Delta^{14}CO_2$ flask samples (10 per month) will be taken at each station selected during the CORSO project, ensuring an even distribution throughout the year to capture seasonal variations. Maintaining a consistent number of samples per station and distributing them evenly throughout the year is essential, as this strategy captures comprehensive temporal




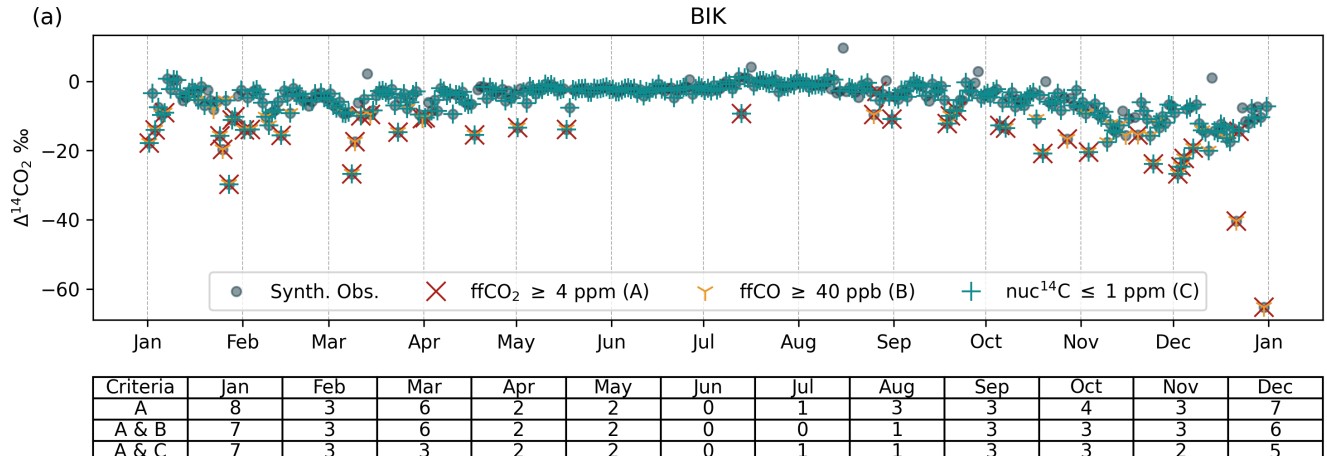

| Criteria | Jan | Feb | Mar | Apr | May | Jun | Jul | Aug | Sep | Oct | Nov | Dec |
|----------|-----|-----|-----|-----|-----|-----|-----|-----|-----|-----|-----|-----|
| A | 8 | 3 | 6 | 2 | 2 | 0 | 1 | 3 | 3 | 4 | 3 | 7 |
| A & B | 7 | 3 | 6 | 2 | 2 | 0 | 0 | 1 | 3 | 3 | 3 | 6 |
| A & C | 7 | 3 | 3 | 2 | 2 | 0 | 1 | 1 | 3 | 3 | 2 | 5 |

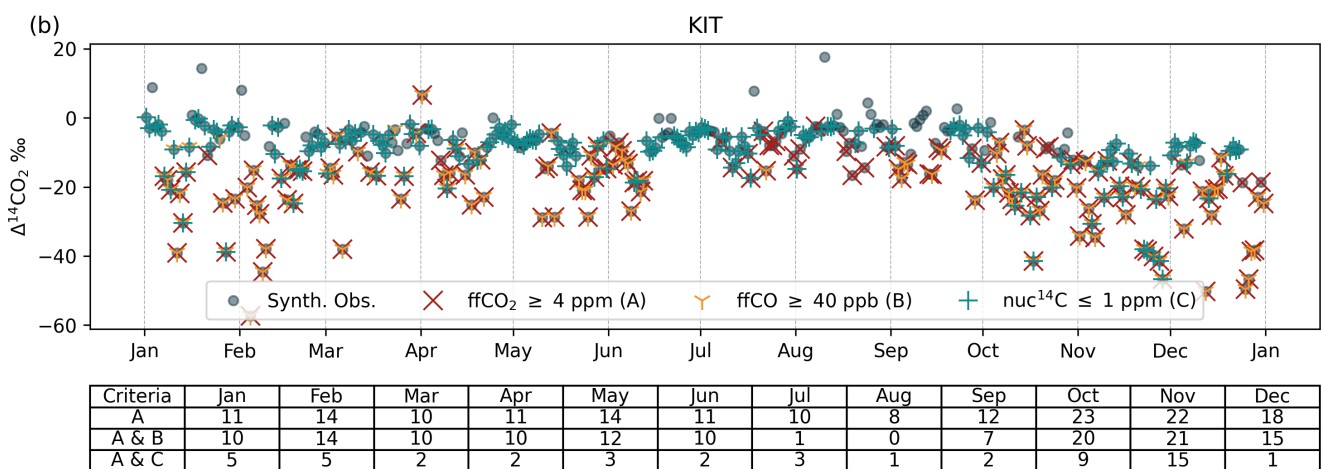

| Criteria | Jan | Feb | Mar | Apr | May | Jun | Jul | Aug | Sep | Oct | Nov | Dec |
|----------|-----|-----|-----|-----|-----|-----|-----|-----|-----|-----|-----|-----|
| A | 11 | 14 | 10 | 11 | 14 | 11 | 10 | 8 | 12 | 23 | 22 | 18 |
| A & B | 10 | 14 | 10 | 10 | 12 | 10 | 1 | 0 | 7 | 20 | 21 | 15 |
| A & C | 5 | 5 | 2 | 2 | 3 | 2 | 3 | 1 | 2 | 9 | 15 | 1 |

**Figure 2.** Synthetic $\Delta^{14}CO_2$ flask samples at a) Białystok (BIK) and b) Karlsruhe (KIT), two of the 12 sampling sites selected for the intensive sampling campaign during the CORSO project. The time series for the remaining sampling sites can be found in Appendix A1. The tables below each figure show the number of synthetic observations per month that meet the $ffCO_2$ threshold (red cross), the $ffCO_2$ and $ffCO$ (yellow tri) thresholds, and the $ffCO_2$ and $nuc^{14}C$ (green cross) thresholds.

coverage and accounts for seasonal changes. For this reason, we focus on selecting the most synthetic samples that meet the criteria in each of the OSSEs describe in the following section and complete the 10 samples per month with synthetic samples that are close to the criteria.





### 3.6 Observing System Simulation Experiments

In the following sections, we describe the experiments. We summarize the setup of the experiments and their criteria in Table 2. As part of the evaluation of the experiments, we calculate the posterior uncertainty of each OSSE with a Monte Carlo ensemble of 25 members.

#### 3.6.1 Base case scenario (BASE)

In the first inversion, BASE, we replicate the current setup of the ICOS network using synthetic $\Delta^{14}CO_2$ integrated samples
and synthetic $CO_2$ observations. In this experiment, we use all stations in Fig. 1 (except MHD, HFD and BIK) and integrated samples according to the column 'Current $\Delta^{14}CO_2$ sampling' in Table 1 (yellow and red dots in Fig. 1). At all stations, we use $CO_2$ observations within the times of the day described in Section 3.3 according to the altitude of the sampling station: midday for lowlands and coastal sites, midnight for mountaintop sites.

#### 3.6.2 Including $\Delta^{14}CO_2$ flask samples (CORSO)

The selection of flask samples represents many logistic and operational challenges. The simulations and data analysis to determine if a sample meets the selection criteria are often conducted weeks after the sample has been taken. As a result, more than 10 samples need to be collected each month, which requires sufficient flasks, storage, and transport capacity. Therefore, we will begin by evaluating the use of synthetic $\Delta^{14}CO_2$ flask samples in the simplest form: taking a sample every 3 days at 13:00 local time, regardless of its composition. This experiment also works as a base case for the use of $\Delta^{14}CO_2$ flask samples. The
selection in this and the following experiments is carried out in the sampling sites marked with yellow dots in Fig. 1. This is the basic approach to sampling selection in the CORSO project when it is not possible to perform near-real-time simulations to estimate the fossil or nuclear contribution of the $\Delta^{14}CO_2$ flask samples.

#### 3.6.3 Applying fossil fuel-related thresholds (ffCO$_2$ and ffCO)

The subsequent experiments are designed following the thresholds described in Sec. 3.3. We do a forward run using the prior
fluxes mimicking a near-real-time simulation, and based on the mixing ratios by flux category, we apply the thresholds to select the synthetic observations. These thresholds are not always met for all 10 observations in a month and in the case of fossil $CO_2$ and CO, during the summer months when fossil emissions are lower, the thresholds are almost never met at most stations, consistent with the seasonal decrease in fossil emissions, as already discussed by Levin et al. (2020). Figure 2 further illustrates the variability in meeting thresholds at different sites.

In months where one of the thresholds or a combination of them is not met, we still need to select the 10 synthetic observations that best fit the experimental conditions. The first experiment including the thresholds is CORSO_ffCO2, where we select synthetic observations at 13:00 LT with a fossil $CO_2$ component greater than or equal to 4 ppm (see Figure 2). Generally, we select the 10 synthetic observations per month with the highest fossil $CO_2$ component. The second experiment is





CORSO_ffCO2_ffCO (criteria A & B in Figure 2). In this experiment, when both thresholds are not met, we select the best
combination with the highest values of ffCO$_2$ and ffCO.

### 3.6.4   Evaluating the impact of nuclear emissions (nuc$^{14}$C)

Here we perform two experiments, one with a low nuclear component (CORSO_ffCO2_nuc14C) and the other with a high
nuclear component (CORSO_ffCO2_nuc14C_max) while ensuring a high ffCO$_2$ composition in both, aiming to evaluate the
impact of the nuclear emissions in the estimation of fossil CO$_2$ emissions. In this way, we intend to completely isolate the
problem by evaluating the impact solely due to nuclear emissions. Similarly to ffCO$_2$ and ffCO the nuclear C$\Delta^{14}$C threshold is
not always met. For instance, synthetic observations at Karlsruhe (KIT) often do not meet the threshold, suggesting significant
nuclear emission influences. This means that if we select the $\Delta^{14}CO_2$ flask samples every 3 days or using only the ffCO$_2$
threshold, there is a high probability of selecting samples with a high nuclear component in sites such as KIT. In contrast,
synthetic observations at Białystok (BIK) frequently meet this threshold, indicating lower nuclear emission impacts (see Fig.
360   2).

In CORSO_ffCO2_nuc14C (criteria A & C in Figure 2), we follow the procedure:

1. We first select the observations that meet both the ffCO$_2$ and nuc$^{14}$C thresholds.

2. For each site, year, and month, we select the top 10 observations with the minimum nuclear influence.

3. If there are less than 10 observations, we fill the remaining slots with observations meeting the ffCO$_2$ thresholds and
moderate nuclear influence (between 1 and 2 ppm C$\Delta^{14}$C).

4. If still short of 10 observations, we fill the remaining slots with observations meeting only the nuclear threshold, ensuring
   the highest possible fossil CO$_2$ influence.

In the CORSO_ffCO2_nuc14C_max experiment, we aim to capture the effect of high nuclear emission events by selecting
observations with the highest nuclear contribution. Initially, we filter for observations that meet the ffCO$_2$ threshold but exclude
those that meet the nuc$^{14}$C threshold. These observations are then sorted by fossil and nuclear emissions, and we select the
top 10 with the highest nuclear emissions for each site, year, and month. If fewer than 10 observations are available, we fill
the remaining slots with observations having fossil emissions between 3 and 4 ppm C$\Delta^{14}$C, still excluding those that meet
the nuc$^{14}$C threshold. If additional observations are needed, we use all remaining observations selecting the ones with highest
fossil contribution.

With these two sets of observations, we perform a separate Monte Carlo ensemble to evaluate the uncertainty in the posterior
fossil CO$_2$ emissions associated only with the influence of nuclear emissions. For this, we randomize the true nuclear emissions
based on an uncertainty equal to the annual budget (0.62 Pg C$\Delta^{14}$C, 100%), recalculate the synthetic observations, and perform
the inversions while maintaining the same setup. This provides a comprehensive assessment of the uncertainties associated with
our experimental setup and the nuclear emissions.





**Table 2.** Summary of the OSSEs performed in this study.

| Simulation | $\Delta^{14}CO_2$ sample type | Criteria |
|---|---|---|
| BASE | Integrated | Current network |
| CORSO | Integrated and Flask | Flask samples at 13LT every third day |
| CORSO_ffCO2 | Integrated and Flask | $ffCO_2 \geq 4ppm$ |
| CORSO_ffCO2_ffCO | Integrated and Flask | $ffCO_2 \geq 4ppm$ & $ffCO \geq 40ppb$ |
| CORSO_ffCO2_nuc14C | Integrated and Flask | $ffCO_2 \geq 4ppm$ & $nuc^{14}C \leq 1ppm$ |
| CORSO_ffCO2_nuc14Cmax | Integrated and Flask | $ffCO_2 \geq 4ppm$ & $nuc^{14}C > 1ppm$ |

## 4  Results

### 4.1  Characterization of the sampling sites in terms of $\Delta^{14}CO_2$

We start by analyzing and comparing the real $\Delta^{14}CO_2$ integrated samples (ICOS RI et al., 2024) with the synthetic observations at the sites selected for the intensive $\Delta^{14}CO_2$ flask sample campaign (Fig. 3 and Appendix A2). Real observations show pronounced seasonal but also episodic fluctuations in $\Delta^{14}CO_2$, such as low values during February and March in CBW ($-16.64‰$), OPE ($-15.14‰$), and KRE ($-5.75‰$) (black line; see Fig. 3), which also coincide with the reduction in modeled observations with FLEXPART (between $-7.6$ in CBW and $-5.1‰$ in KRE, teal line) and can be associated with the typically high fossil emissions during winter. On the other hand, there are also some high values during January and February in KRE ($2.56‰$) and OPE ($6.25‰$). These elevated values are primarily related to nuclear emission enrichment. However, during the growing season, when heterotrophic respiration is more active than in winter, these values can also be influenced by isotopic disequilibrium. This disequilibrium is a consequence of the radiocarbon absorbed by the biosphere during the period of elevated atmospheric $^{14}C$ levels following nuclear weapon tests conducted between 1945 and 1980. When this carbon is released through heterotrophic respiration, it typically has a $\Delta^{14}C$ signature that is higher than that of the current atmosphere.

Although the synthetic observations are calculated with non-optimized fluxes, we find certain reproducibility of the seasonal patterns at sites such as CBW where we have the best agreement between the real and synthetic observations with the highest correlation coefficient (R) and lowest mean bias deviation (MBE) (see Fig. 3), and KRE and SAC (Fig. A2) in which the synthetic observations mostly underestimate the real observations (negative MBD). Also, at some sampling stations such as JFJ and PAL, the synthetic observations do not capture the variability shown by the real observations, and it is reflected in high root mean square error values (RMSE, see Fig. A2).





**Figure 3.** Comparison of the available real $\Delta^{14}CO_2$ integrated samples (black) (ICOS RI et al., 2024) with the modeled background observations (red) and synthetic observations (teal) at a) CBW, b) KRE and c) OPE, three of the sampling sites selected for the intensive sampling campaign during the CORSO project.





## 4.2 OSSEs

We evaluate the retrieval of fossil $CO_2$ emissions by comparing the assumed true values derived from EDGAR against the prior estimates from ODIAC and the posterior estimates of the experiments described in Section 3.6. In this section, we focus the analysis on the bias and uncertainty reduction calculated as follows:

$$\text{Bias reduction} = \left( 1 - \frac{|\text{Posterior} - \text{Truth}|}{|\text{Prior} - \text{Truth}|} \right) \times 100 \tag{4a}$$

$$\text{Uncertainty reduction} = \left( 1 - \frac{\text{Posterior uncertainty}}{\text{Prior uncertainty}} \right) \times 100 \tag{4b}$$

### 4.2.1 Impact of adding $\Delta^{14}CO_2$ flask samples

In the study domain (Figure 4a), the true emissions show a seasonal variation with peaks in winter and troughs in summer, reaching a peak of 4.79 TgC day$^{-1}$ in January and reducing to a minimum of 3.13 TgC day$^{-1}$ in July. The prior estimates significantly underestimate the true emissions, with a bias as large as 29% in January and greater than 12% on an annual
basis, with a minimum of 7% and 1% in June and July, respectively. In general, there is a larger bias reduction in the CORSO experiment with the exception of June and July, where the emissions are overestimated, with values between 22% in October and 98% in May. The BASE experiment shows a better agreement for June and July, but a lower bias reduction for October and May, with 4% and 73%, respectively. The prior uncertainty for the study domain ranges from 50% in January to 72% in August. The uncertainty reduction is similar in both experiments, with values ranging from 71% to 87% for CORSO, and from
71% to 75% for BASE.

WCE and Germany, where around 30% and 16% of the total emissions occur, respectively, have similar results in relative terms. Both regions have a larger prior bias during winter, with the largest biases occurring in January (35% for WCE and 42% for Germany). In contrast, they exhibit a lower bias in summer, with a minimum in July (5% for WCE and 4% for Germany). The posterior emissions of both experiments overestimate the monthly budgets during summer, from June to August in WCE
and from May to August in Germany. However, the CORSO experiment shows values closer to the truth in this season. Outside of the summer season, the BASE experiment demonstrates a larger bias reduction in WCE, whereas the CORSO experiment shows a larger bias reduction in Germany. The prior uncertainties in both regions exceed 100% but are consistently reduced by more than 90% by the CORSO experiment in both WCE and Germany, and by more than 80% by the BASE experiment. Nevertheless, from May to September the absolute posterior uncertainty of both experiments in both regions is larger than their
respective absolute prior bias.

France, the Benelux region and the British Isles have similar monthly budgets in magnitude, with values between 0.2 and 0.4 TgC days$^{-1}$, and similar prior biases between 5% (mainly for Benelux) and 7%. The posterior estimates of both experiments are similar in France and the Benelux, with some months (May to June) having good agreement (bias reduction greater than 50%). In the case of the British Isles, there are larger differences between the posterior estimates for both experiments and less





**Figure 4.** Monthly fossil $CO_2$ truth (black dashed lines), prior (red dotted lines), and posterior fluxes from the BASE (teal solid lines) and CORSO (yellow solid lines) experiments for a) the study domain and 5 sub-regions : b) Western/Central Europe, c) Germany, d) France, e) Benelux, and f) British Isles. The shaded areas represent the uncertainty ($1\sigma$) calculated in a Monte Carlo ensemble of 25 members.





occurrences of months with bias reduction greater than $50\%$. With regard to the posterior uncertainties, the largest reductions are found in the Benelux with similar values for both experiments between $80\%$ and $90\%$. In France and the British Isles, there is a larger uncertainty reduction from the CORSO experiment, with an average of $54\%$ and $70\%$, respectively.

### 4.2.2 Impact of selecting $\Delta^{14}CO_2$ flask samples using the ffCO$_2$ and ffCO thresholds

Here, we compare the CORSO_ffCO2 and CORSO_ffCO2_ffCO experiments against the base case, CORSO, to evaluate the
impact of selecting $\Delta^{14}CO_2$ flask samples using the ffCO$_2$ and ffCO thresholds. This time we focus only on Western/Central Europe and Germany (see Fig. 5), which show the best results in Section 4.2.1.

 In Western/Central Europe (WCE), the CORSO experiment generally shows a bias reduction of between $81\%$ and $98\%$ in winter with an uncertainty reduction of $82\%$ to $91\%$. The CORSO_ffCO2 experiment achieves a bias reduction of $71\%$ to $99\%$ in winter and an uncertainty reduction of $89\%$ to $92\%$, similar to the CORSO_ffCO2_ffCO experiment with a bias reduction of
$79\%$ to $97\%$ and an uncertainty reduction of $81\%$ to $94\%$ for the same period. During the summer, CORSO_ffCO2_ffCO has the best bias reduction in July ($78\%$ vs. $-42\%$ and $15\%$ from CORSO and CORSO_ffCO2, respectively), while CORSO and CORSO_ffCO2 have a better recovery in June and August (between $70\%$ and $88\%$ vs. $26\%$ to $56\%$ from CORSO_ffCO2_ffCO). The uncertainty reduction is similar for all three experiments through the year with values greater than $79\%$.

 Likewise, in Germany, the uncertainty reduction for the three experiments is greater than $83\%$ throughout the year. The
largest differences in bias reduction occur between May and August. CORSO_ffCO2 shows the best results during this period, with a bias reduction between $48\%$ in July and $97\%$ in June, while the other two experiments show values as low as $4\%$ in August for CORSO and $7\%$ in July for CORSO_ffCO2_ffCO. The latter, in general, shows the lowest reduction in bias during summer, with a maximum reduction of $56\%$ in June.

### 4.2.3 Impact of nuclear power facilities

In Figure 6, we show the emission time series for the CORSO_ffCO2_nuc14C and CORSO_ffCO2_nuc14Cmax experiments and the posterior uncertainties resulting from the Monte Carlo ensemble performed to evaluate the impact of nuclear emissions. The time series of the two experiments are very similar and close to the true emissions. However, the uncertainty of the posterior CORSO_ffCO2_nuc14C fossil CO$_2$ emissions is consistently lower throughout the year, ranging from $12\%$ to $44\%$ in WCE and from $6\%$ to $17\%$ in Germany, compared to the CORSO_ffCO2_nuc14Cmax experiment (from $42\%$ to $118\%$ in WCE
and $11\%$ to $57\%$ in Germany). The uncertainty reduction for the CORSO_ffCO2_nuc14C experiment is high, with more than $80\%$ for all months in WCE, and over $88\%$ in Germany. On the other hand, CORSO_ffCO2_nuc14Cmax shows the lowest uncertainty reduction from July ($38\%$) to September ($55\%$) in WCE and from August ($70\%$) to October ($71\%$) in Germany.

 The spatial distribution of the annually aggregated prior and posterior uncertainties and the corresponding uncertainty reduction, are shown in Figure 7. The prior uncertainty highlights regions with high initial uncertainty (darker purple shades, Fig.
7a), such as parts of western Europe around the Benelux region and the northern part of France and Germany, as well as England where the nuclear facilities with the highest emissions are also located (Fig. 7b), such as the spent fuel reprocessing plants of La Hague (FR) and Sellafield (EN). The posterior CORSO_ffCO2_nuc14C shows significant reductions in uncertainty in




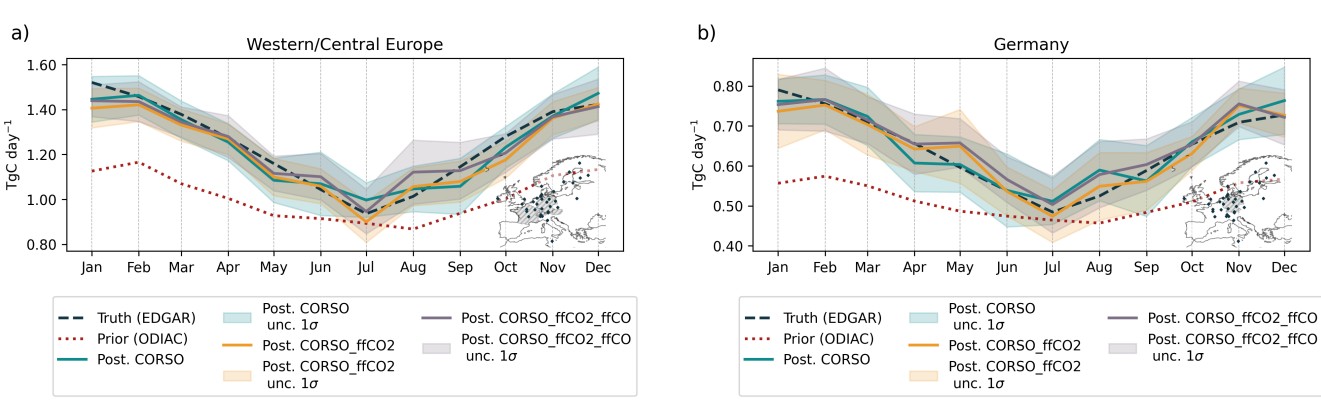

**Figure 5.** Monthly fossil $CO_2$ truth (black dashed lines), prior (red dotted lines), and posterior fluxes from the CORSO (teal solid lines) and CORSO_ffCO2 (yellow solid lines) and CORSO_ffCO2_ffCO (purple solid lines) experiments for a) Western/Central Europe and b) Germany. The shaded areas represent the uncertainty ($1\sigma$) calculated in a Monte Carlo ensemble of 25 members. We excluded the prior uncertainty for a better visualization and is the same as panels b) and c) in Fig. 4.

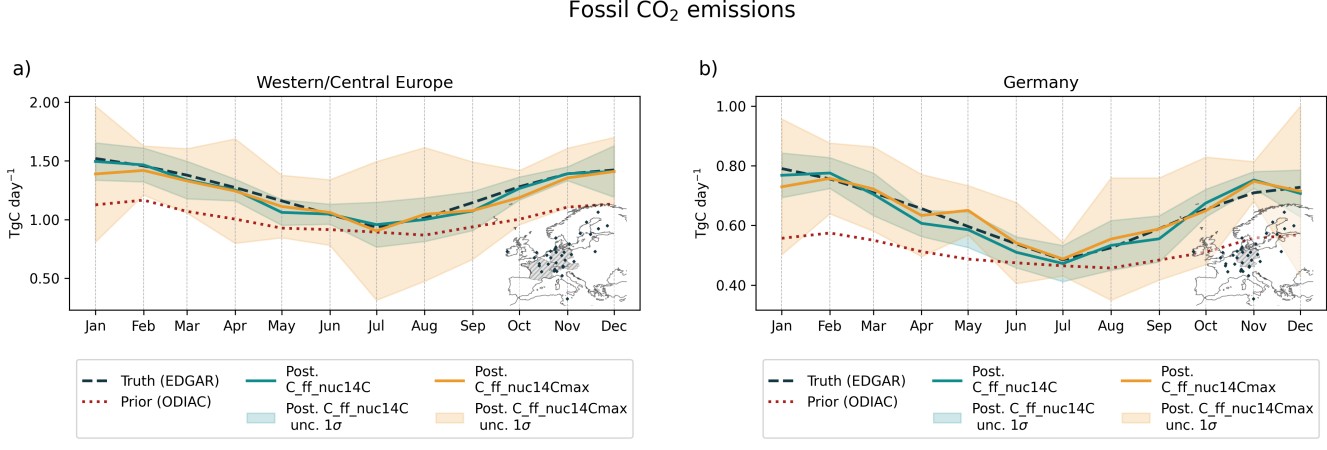

**Figure 6.** Monthly fossil $CO_2$ emissions for the average of the posterior values from the CORSO_ffCO2_nuc14C (teal) and CORSO_ffCO2_nuc14Cmax (yellow) experiments in comparison with the truth (black-dashed) and prior (red-dotted) values for a) Western/Central Europe and b) Germany. The shaded areas show the posterior uncertainties ($1\sigma$) for the two experiments.



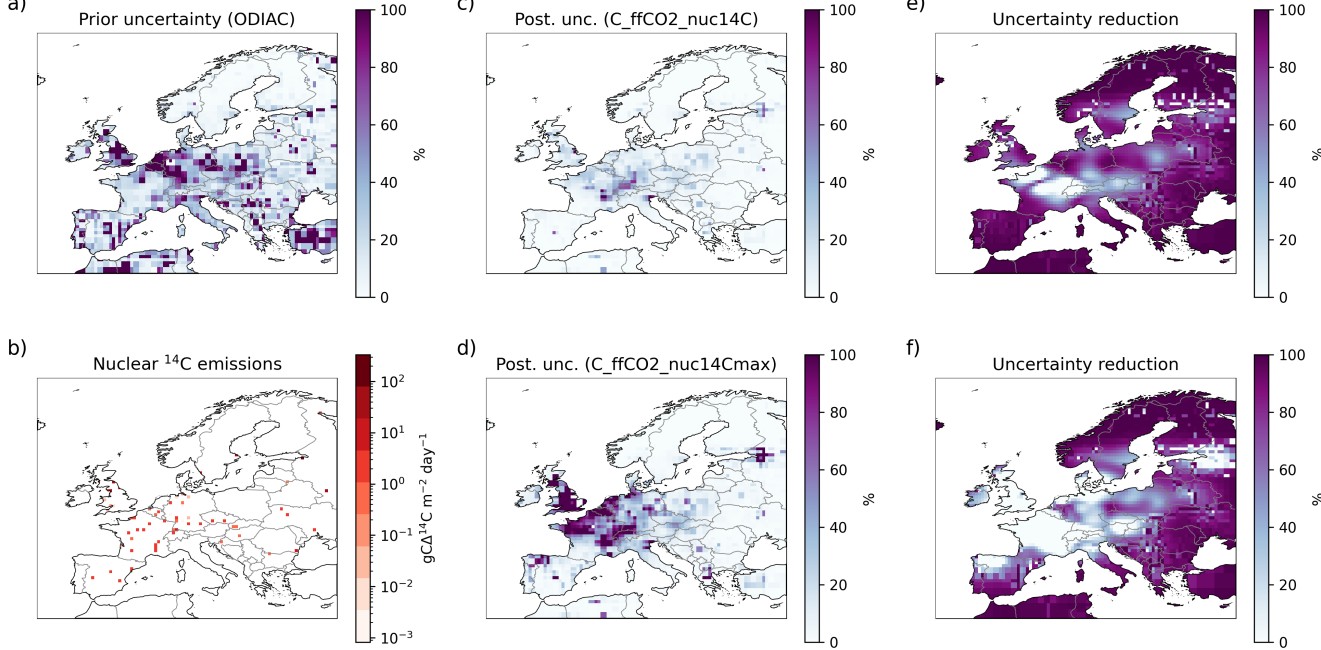

**Figure 7.** Spatial distribution of the annual a) prior fossil $CO_2$ emission uncertainty, b) nuclear radiocarbon emissions, posterior uncertainties of the c) CORSO_ffCO2_nuc14C and d) CORSO_ffCO2_nuc14Cmax experiments, and their respective uncertainty reductions, panels e) and f).

most of Europe (Fig. 7c). The dark purple areas in Figures 7e, particularly in Western and Central Europe, indicate the regions with the highest uncertainty reductions. For example, large uncertainty reductions are observed in Benelux and east Germany,

east France, and west England, with uncertainty reductions often exceeding the 70% per grid cell. However, there is very low uncertainty reduction in the north and the southeast of France, covering also Switzerland. The CORSO_ffCO2_nuc14Cmax experiment, which focuses on selecting $\Delta^{14}CO_2$ flask samples with high nuclear emissions but still in compliance with the ffCO$_2$ threshold, also achieves substantial uncertainty reductions in some areas where nuclear emissions are low or even absent (Fig. 7f). However, the low uncertainty reduction that we found in France for the CORSO_ffCO2_nuc14C experiment (21%

aggregated over the country) spreads over most of Switzerland, France, the British Isles (mainly England) and Denmark for the CORSO_ffCO2_nuc14Cmax experiment. In this experiment, the uncertainties increased by 328% in Switzerland, 115% in France, 78% in the British Isles and 61% in Denmark. For the CORSO_ffCO2_nuc14C in Switzerland, there is also an increase in the posterior uncertainties of 100%.





## 5   Discussion

The real integrated $\Delta^{14}CO_2$ samples provide evidence of high nuclear emissions, particularly during periods outside the growing season, when we have high fossil $CO_2$ emissions and low heterotrophic respiration (i.e. low terrestrial isotopic disequilibrium) and would therefore expect negative $\Delta^{14}CO_2$ values. For example, elevated $\Delta^{14}CO_2$ values observed around May in Karlsruhe (KIT, DE), between January and February in Křešín u Pacova (KRE, CZ), and March in Hohenpeißenberg (HPB, DE) are indicative of significant nuclear emissions. These observations are consistent with the findings of Bozhinova et al.

(2014), who demonstrated that nuclear emissions could significantly alter the $\Delta^{14}CO_2$ signature, complicating the differentiation between fossil fuel-derived and naturally occurring $CO_2$. Maier et al. (2023) further emphasized the challenges posed by nuclear emissions, highlighting that in regions with high nuclear activity, radiocarbon emissions from nuclear facilities could mask the fossil fuel signal in $\Delta^{14}CO_2$ measurements. This masking effect leads to potential biases in $CO_2$ attribution, complicating efforts to isolate fossil $CO_2$ from other sources. To address this issue, Maier et al. (2023) underscored the need to

incorporate high-resolution nuclear emission data into atmospheric models to correct for these biases. In this study, we simulate the intensive $\Delta^{14}CO_2$ flask sampling campaign during the CORSO project in different scenarios to address the challenge of using atmospheric $\Delta^{14}CO_2$ measurements for the estimation of fossil $CO_2$ emissions in Europe, a continent with a high concentration of active nuclear facilities. We address this challenge by evaluating the impact of taking more samples and for shorter periods (1 hour instead of 14 days) throughout the year, allowing us to capture regions and periods with high fossil

emissions and low nuclear emissions.

In our first experiment, we study the impact of $\Delta^{14}CO_2$ flask samples on the estimation of fossil $CO_2$ emissions by comparing the BASE experiment (using only integrated samples) with the CORSO experiment (including additional flask samples). Our findings reveal that, in general, the CORSO experiment provides a better estimation of emissions, particularly in winter months, and significantly reduces both bias and uncertainty compared to the BASE experiment. In the study domain, the

CORSO experiment has a larger bias reduction and uncertainty reduction throughout most months, except for June and July, where the BASE experiment performs better. June and July are the months with the lowest fossil emissions of the year, as already found by Levin et al. (2020) in real observations and in this study with synthetic observations. Levin et al. (2020) found that fossil $CO_2$ events are particularly rare during the summer months, with very few significant events occurring between May and August. In these months, fossil $CO_2$ mixing ratios rarely exceeded 4-5 ppm at various stations. Since integrated samples

cover longer periods and hence larger areas than flask samples, they are more likely to capture the cumulative effects of low but steady emissions over time, providing a better estimate during months with fewer significant fossil $CO_2$ events. This extended sampling period compensates for the lower frequency of elevated emissions, ensuring that even minor contributions are accounted for, which may explain the improved performance of the BASE experiment during the summer months. However, in some subregions, such as Western/Central Europe (WCE) and Germany, the CORSO experiment shows a better estimate

of $CO_2$ emissions and a larger uncertainty reduction throughout the year, compared to BASE. Despite this improvement, both regions still experience relatively high posterior uncertainties during the summer months. In France, Benelux, and the British





Isles, the CORSO experiment consistently outperforms the BASE experiment in reducing uncertainty, particularly in the British Isles, where uncertainty reduction reaches an average of $70\%$.

As already stated by Levin et al. (2020), it is necessary to perform a sample selection of $\Delta^{14}CO_2$ flask samples to ensure a
good constraint on fossil $CO_2$ emissions, based on the thresholds defined for $CO_2$ and CO. This approach helps to guarantee the detection limit of the $\Delta^{14}CO_2$ analysis, isolate fossil $CO_2$ signals from other sources of $CO_2$ and make a more efficient use of flask samples. However, this method also carries the risk of predominantly monitoring the same dominant point sources, which may not represent a comprehensive mixture for the region. To mitigate this risk, it is essential to balance the selection criteria to capture a more representative mix of regional sources. Furthermore, the uncertainty of the $\Delta^{14}CO_2$ analysis requires
a minimum signal strength to ensure the accuracy of the measurements. This requires the inclusion of samples that meet the fossil contamination thresholds and provide a sufficient radiocarbon signal to reduce the analytical uncertainty. Ensuring a minimum signal strength is crucial for the reliability of the $\Delta^{14}CO_2$ data, as low signal samples can lead to higher relative errors and less confidence in fossil $CO_2$ estimates.

We applied the thresholds proposed by Levin et al. (2020) in two experiments, one applying $ffCO_2 \geq 4ppm$ (CORSO_ffCO2),
and the other combining it with $ffCO \geq 40ppb$ (CORSO_ffCO2_ffCO), and compare the results against the CORSO experiment, focusing on Western/Central Europe (WCE) and Germany. The findings reveal that all three experiments achieve significant reductions in bias and uncertainty, particularly during winter. In WCE, bias reductions range from $81\%$ to $98\%$ in the CORSO experiment, with similar reductions in the other two experiments. However, during summer, CORSO_ffCO2_ffCO shows a better bias reduction in July, while CORSO and CORSO_ffCO2 perform better in June and August. The reduction
in uncertainty remains high in all experiments, with values consistently higher than $79\%$. In Germany, the results are similar, with all experiments showing a high uncertainty reduction and the largest differences in bias reduction occurring during the summer months. CORSO_ffCO2 generally shows the best results during this period, while CORSO_ffCO2_ffCO tends to have the lowest bias reduction during the summer. Nevertheless, the $ffCO_2$ and ffCO thresholds are based on suggested values and should be adapted at each station according to their specific $ffCO/ffCO_2$ ratio or $CO/CO_2$ emission ratio. If the emission
ratio would result in a ratio of $40ppb/4ppm$, the results of experiments CORSO_ffCO2 and CORSO_ffCO2_ffCO would be identical.

The analysis of the experiments shows that there is not a single experiment that consistently outperforms the others across all seasons and regions. Although each approach (CORSO, CORSO_ffCO2, and CORSO_ffCO2_ffCO) offers its own strengths in bias and uncertainty reduction, particularly during the winter months, none stands out as consistently better across all
scenarios. Implementing such a sampling strategy in a real-world operational setting would require performing near-real-time simulations to estimate the $ffCO_2$ component. The CO threshold was introduced because this can be obtained from continuous CO measurements, and can be calculated as the CO enhancement with respect to the background instead of ffCO (Levin et al., 2020). From the perspective of Observing System Simulation Experiments (OSSEs), the results suggest that the selection of samples may not be as critical as ensuring a good coverage of sampling events throughout the year. The findings indicate that a
well-distributed and frequent sampling strategy might be more effective in capturing the necessary data for accurate fossil $CO_2$



emission estimates, rather than relying heavily on stringent selection criteria. However, this is not the case when we consider the selection of samples according to their nuclear contamination.

In Europe, with more than 170 operational reactors and two reprocessing plants, nuclear contamination significantly impacts $\Delta^{14}CO_2$ samples. Maier et al. (2023) highlight that the median nuclear contamination at ICOS sites accounts for about 30%
in day-and-night integrated samples and 15% in midday integrated samples, leading to substantial underestimation of fossil $CO_2$ estimates if not corrected. Similarly, Graven and Gruber (2011) discuss the continental-scale enrichment of atmospheric $\Delta^{14}CO_2$ due to emissions from the nuclear power industry, which creates significant gradients that extend more than 700km from nuclear sites in Europe. Their study demonstrates that the spatial scale of these gradients is sufficient to influence regional $\Delta^{14}CO_2$ levels, requiring high-resolution data from each nuclear reactor to accurately estimate $\Delta^{14}CO_2$ enrichment
and mitigate biases in fossil $CO_2$ estimates (Graven and Gruber, 2011).

Here, we investigate the impact of nuclear emissions by performing two experiments, both following the ffCO$_2$ selection threshold, but one selecting samples with low nuclear contamination (CORSO_ffCO2_nuc14C), and the other selecting samples with high nuclear contamination (CORSO_ffCO2_nuc14Cmax). The findings show that while both experiments produce similar emission time series that closely align with the true emissions, the uncertainty in the CORSO_ffCO2_nuc14C experi-
ment is consistently lower throughout the year. The CORSO_ffCO2_nuc14C experiment achieves substantial uncertainty reductions, with reductions exceeding 80% in all months in WCE and 88% in Germany. In contrast, the CORSO_ffCO2_nuc14Cmax experiment shows higher uncertainty, particularly during the summer months. Spatial analysis reveals that the CORSO_ffCO2_nuc14C experiment significantly reduces uncertainty across most of Europe, especially in regions with high prior uncertainty, such as Benelux, eastern Germany, eastern France and western England. However, countries with high nuclear emissions or regions
surrounded by high nuclear emissions, such as Switzerland, France, England, and Denmark, show low uncertainty reduction in both experiments. In Switzerland, there is even a 100% increase in the posterior uncertainty when selecting samples with low nuclear contamination.

This is a very particular case in which we might see the extended effect of nuclear sites described by Graven and Gruber (2011). In Switzerland, we only have the Jungfraujoch (JFJ) sampling station, which primarily takes integrated samples during
the CORSO sampling campaign. In our simulations, JFJ is sampling emissions from nuclear facilities in Switzerland and also in neighboring Germany and France. The synthetic $\Delta^{14}CO_2$ integrated samples at this station presented enhancements due to nuclear emissions ranging from 0.04‰ to 3.28‰. The latter, corresponding to a synthetic integrated sample starting on 30 July (Fig. 8), presented total $\Delta^{14}CO_2$ values as large as 20‰ during the Monte Carlo ensemble, mostly due to the nuclear emissions. This proximity to nuclear sites might have an impact on the estimation of fossil $CO_2$, leading to the observed
increase in posterior uncertainty. Therefore, despite the general benefits of integrated samples in capturing long-term emission trends, in regions like Switzerland, where nuclear contamination can be a significant factor, the effectiveness of these samples can be compromised. This underscores the importance of balancing the advantages of integrated samples with the need for additional strategies to address nuclear contamination. Although the BASE experiment using only integrated samples shows some strengths in terms of uncertainty and bias reduction in periods of low fossil emissions, the utility of these samples in
regions with high or surrounding nuclear emissions may be limited. This suggests that while integrated sampling can provide







**Figure 8.** Integrated footprint at the Jungfraujoch (JFJ) sampling station on 30 July. Crosses indicate the location of nuclear facilities in the study domain.

a solid baseline, regions affected by nuclear facilities may require a more refined approach that combines integrated and flask sampling or even performs a selection of integrated samples to achieve reliable fossil estimates $CO_2$.

In our perfect transport OSSEs implementation, we do not account for uncertainties due to transport model representation errors. Munassar et al. (2023) found that the use of different transport models, which help us to understand the model represen-

tation error, can result in differences in the posterior carbon budget up to $60\%$. Their study uses continuous $CO_2$ observations selected at times when there is a better model representation. We assume that these discrepancies and in general the model representation of integrated samples could be even worse, since the samples are continuously integrated for 2 weeks. Maier et al. (2022) study the performance of two modeling approaches using a Lagrangian model (STILT) in representing afternoon and nighttime 2-week integrated $^{14}$C-based ffCO$_2$ observations from Heidelberg. Their standard surface source influence (SSI)





approach, similar to our approach with FLEXPART in which all emissions are assumed to occur at ground level, was almost twice better at representing integrated afternoon samples than night-time samples, when comparing modeled and observed ffCO$_2$ mixing ratios in terms of root mean square deviation (RMSD). Heidelberg is a sampling station located at 113 m.a.s.l., therefore it is expected that the models represent better the afternoon conditions when the planetary boundary layer (PBL) is well developed. They propose the volume source influence (VSI) approach in which there is a representation of the emission height and the plume rise of point source emissions, such as the emissions from power plants. For this approach, the performance is similar for the afternoon and night samples. Exploring the model representation at the different sites, and the potential implementation of the volume source influence (VSI) approach proposed by Maier et al. (2022) to reduce the representation error under sampling conditions during an unstable atmosphere, is important for the use of integrated samples in an inversion study using real data.

The definition and characterization of the prior uncertainty is an additional limitation of our study, and this challenge may also be reflected in the posterior uncertainties observed in our experiments. A common outcome across all experiments is that, particularly during the summer months, the posterior uncertainty remains larger than the prior bias. This indicates that despite the improvements achieved through various sampling strategies, we are not yet at the point where we can reliably use estimated monthly emissions for precise fossil CO$_2$ assessments. The persistent high uncertainties during the summer underscore the need for further refinement in both the sampling strategies and the characterization of uncertainties, as well as in the inverse modeling approach itself. Enhancing the accuracy and robustness of these models is essential to better capture the complexities of fossil CO$_2$ emissions, especially in seasons where the signal is weaker and more susceptible to variability. Until these challenges are addressed, the utility of monthly emissions estimates will remain limited, highlighting the importance of ongoing research and development in this area.

## 6 Conclusions

In this study, we find that adding regular $\Delta^{14}CO_2$ flask sampling to the integrated sampling (CORSO) generally provides better emission estimates than using only integrated samples (BASE), particularly during the winter months. However, the BASE experiment performed better than CORSO during low-emission months such as June and July. We also find that the selection of synthetic $\Delta^{14}CO_2$ flask samples according to their fossil contribution did not show significant improvements compared to the simpler CORSO approach. However, when samples were selected according to their level of nuclear contamination, the experiments showed that selecting samples with low nuclear contamination led to a substantial reduction in uncertainty, particularly in regions like Western/Central Europe and Germany. In contrast, selecting samples with high nuclear contamination resulted in higher uncertainties, especially during the summer months.

Therefore, we recommend focusing particularly on the selection of $\Delta^{14}CO_2$ flask samples according to their nuclear contamination given the currently unknown temporal profile of the $\Delta^{14}CO_2$ from most of the nuclear facilities in our model domain. It is also necessary to perform a site-specific revision of the CO, ffCO$_2$, and nuc$^{14}$C thresholds to adjust these values to the intensity of the fluxes measured at each station. This is also important for the $\Delta^{14}CO_2$ integrated samples. Although



they can help to better estimate fossil $CO_2$ in periods of low emissions such as summer, long integration times can also help to capture large $^{14}$C nuclear emissions, which increases the posterior uncertainty of the estimates. In real inversions, these

integrated samples can also have large representation errors. A promising approach to account for these representation error in an inversion is the implementation of the volume source influence (VSI) approach as proposed by Maier et al. (2022).

Despite the advancements shown by these experiments, high posterior uncertainties during the summer months remain a challenge. This limits the reliability of monthly emission estimates, underscoring the need for further refinement in both sampling strategies and inverse modeling techniques. Until these challenges are adequately addressed, the utility of monthly

emissions estimates will remain limited, pointing to the importance of performing an appropriate uncertainty characterization of fossil emissions.

*Code availability.* The LUMIA source code used in this paper has been published on Zenodo and can be accessed at https://doi.org/10.5281/zenodo.8426217

*Data availability.* The data and the scripts used to generate the figures are available at https://github.com/cdgomezo/assets-corso-campaign.
git

## Appendix A: Observations

### A1 Thresholds

### A2 Comparison between real and modeled observations

*Author contributions.* All authors contributed with the design of the experiments, CG and GM developed the code, and CG performed the
simulations. CG prepared the paper, and GM, UK, and MS provided corrections and suggestions for improvements.

*Competing interests.* The authors declare that they have no conflict of interest.

*Acknowledgements.* We thank the Swedish Research Council for Sustainable Development FORMAS for funding the 14C-FFDAS project (Dnr 2018-01771). We acknowledge support from the EU projects AVENGERS (Grant Agreement (GA): 101081322) and CORSO (GA: 101082194) as well as from the three Swedish strategic research areas ModElling the Regional and Global Earth system (MERGE), the
e-science collaboration (eSSENCE), and Biodiversity and Ecosystems in a Changing Climate (BECC). The computations were enabled by





**Figure A1.** Synthetic $\Delta^{14}CO_2$ flask samples at the 10 remaining sampling sites selected for the intensive sampling campaign during the CORSO project. The tables below each figure show the number of synthetic observations per month that meet the $ffCO_2$ threshold (red cross), the $ffCO_2$ and $ffCO$ (yellow tri) thresholds, and the $ffCO_2$ and $nuc^{14}C$ (green cross) thresholds





**Figure A2.** Comparison of the available real $\Delta^{14}CO_2$ integrated samples (black) (ICOS RI et al., 2024) with the modeled background observations (red) and synthetic observations (teal) at ten ICOS sites.





resources provided by the National Academic Infrastructure for Supercomputing in Sweden (NAISS), the Swedish National Infrastructure for Computing (SNIC) at LUNARC, and NSC partially funded by the Swedish Research Council through grant agreements no. 2022-06725 and no. 2018-05973, and the Royal Physiographic Society of Lund through Endowments for the Natural Sciences, Medicine, and Technology - Geoscience. A special acknowledgment is given to Frank-Thomas Koch and Christoph Gerbig at the Max Planck Institute for Biogeochemistry Jena for producing and providing the fossil CO emissions product, Sourish Basu at the NASA Goddard Space Flight Center for providing the optimized fluxes used for the calculation of the background mixing ratios, and Ida Storm at the ICOS Carbon Portal for providing the annual emissions from nuclear facilities.




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
