# Peer review of "Modeling support for an extensive $\Delta^{14}CO_2$ flask sample monitoring campaign over Europe to constrain fossil $CO_2$ emissions"

_EGUsphere, 2024_

## Author Response (AR1)

Lund, July 18$^{th}$, 2025

**Carlos Gómez-Ortiz**
**Department of Physical Geography and Ecosystem Science**
**Lund University**
**Sweden**

Editorial board
**Atmospheric Chemistry and Physics (ACP)**

We thank both referees for their insightful comments and constructive suggestions. They have contributed to significantly improving the manuscript and its discussion. In the following, we address their comments point-by-point. We use text in italics to repeat the referees' comments, normal text for our response. In addition to responding to the specific comments, we have also revised several parts of the manuscript to improve clarity, flow, and overall readability. Following the suggestion of Referee #1, we have also modified the title to: "Modeling support for an extensive $\Delta^{14}CO_2$ flask sample monitoring campaign over Europe to constrain fossil $CO_2$ emissions."

Please read below our reply to the comments from Referee #1 and Referee #2.

**RC1: 'Comment on egusphere-2024-3013', Dylan Geissbühler**

*General comments: Overall quality of manuscript*

*The study investigates the potential impact of an intensive Δ¹⁴CO₂ sampling campaign across Europe as part of the CORSO project for an intended 2024 campaign, based on 2018 data across 10 ICOS network stations. By integrating these measurements with CO₂ data in inverse modeling, the research aims to enhance fossil CO₂ emission estimates. The authors use Observing System Simulation Experiments (OSSEs) to assess 3 different sampling strategies: uniform sampling every 3rd day, samples with a focus on high fossil fuel CO₂ contributions and samples with reduced nuclear ¹⁴C contamination. The results suggest that a denser sampling network and targeted selection strategies significantly reduce uncertainty in fossil fuel emission estimates, particularly in regions with nuclear influence, focusing on high fossil CO2 contamination could improve accuracy of emissions estimates, and considering nuclear influence could help minimize potential biases. The study highlights the importance of sample selection for 14CO2 analysis.*

*In general, the study is well constructed and the hypotheses are clearly formulated making it easy to follow. However some elements could still be improved:*

*The paper is quite long and a few passages in the text are essentially repetitions from one another, 2 examples: Nuclear emissions of 14C are explained in the introduction (lines 28 – 34 and detailed after) but then also in section 3.2 (238 – 241), and at the start of the discussion. Also the terrestrial disequilibrium due to the nuclear bomb testing is mentioned multiple times as well (section 2, section 3.1, and explained again fully in section 4.1). Check the paper in general for conciseness. A mention of how a subsampling of flask is especially relevant for 14CO2 measurements (cost, sample processing) is missing, in my opinion.*

*There are some consistency problems in notations: 14C- radiocarbon, spaces before units, Fig. Vs Figure,...) See below for some examples, but check manuscript thoroughly for these kinds of things.*

R/ We thank the referee for their thorough and constructive comments.

- Regarding the length and repetition: We have carefully revised the manuscript to remove redundancies and improve conciseness, particularly in sections discussing nuclear emissions and terrestrial disequilibrium.
- On the relevance of flask subsampling for $\Delta^{14}CO_2$ measurements: We have added a sentence in the introduction (Section 1) to highlight the high cost and complexity associated with radiocarbon analysis, which underscores the importance of efficient sampling strategies.
- On notation and formatting consistency: We have reviewed the manuscript thoroughly and corrected inconsistencies in the use of terms (e.g., $^{14}C$, radiocarbon), spacing before units, and formatting of figure references to ensure a consistent and professional presentation throughout.

*Specific comments:*

*Line 14: are we really talking about a sampling strategy, or more of a subsampling strategy?*

R/ The sample selection occurs after the sample is taken, through near-real-time simulations and before it is analyzed. In this sense, "subsampling strategy" might indeed be a more accurate term. However, to maintain clarity and avoid introducing new terminology that could raise additional concerns, we have decided to replace the term "sampling strategy" and its variations with "flask sample selection strategy", or simply "selection strategy" throughout the paper.

*Line 26: check consistency of 14C vs radiocarbon used throughout the manuscript (goes from one to the other)*

R/ We have thoroughly checked the manuscript for consistency and ensured that "radiocarbon" is used in the text when referring to $^{14}C$ and $^{14}CO_2$, reserving the chemical notation for formulas, figures, and parenthetical references.

*Line 37: datasets instead of data sets*

R/ This and other occurrences have been corrected throughout the manuscript.

*Line 50: remove "such as for Europe"*

R/ We have removed "such as for Europe" to improve the clarity and flow of the sentence, as the full list of regions (Europe, North America, and East Asia) already sufficiently illustrates the point without needing a specific example.

*Line 52: "The research" -> "Research"*

R/ We have removed "The" to improve clarity.

*Lines 54 to 56: this should be moved to previous paragraph*

R/ We have moved the sentence to the previous paragraph.

*Line 60: space before m unit*

R/ We have carefully reviewed the manuscript and ensured that all units, including the use of proper spacing between numerical values and unit symbols (e.g., 100~\unit{m}), are consistently typeset following the Copernicus and IUPAC standards throughout the manuscript.

*Line 61: the network stations -> the network of stations*

R/ We have revised the sentence to simply refer to "the network," which improves clarity and conciseness while keeping the meaning clear.

*Line 62: "the measurements represent" -> "measurements should represent"*

R/ We have revised it to state that the measurements are "intended to represent" large areas, which better reflects the design goal of the network while improving the overall clarity and precision of the sentence.

*Line 66: "these stations" which stations are we talking about, all of them?*

R/ We have modified the text to avoid ambiguities as follows:

"Since 2015, an increasing number of ICOS stations have been collecting 1-hour flask samples regularly. Of the approximately 100 flask samples taken per station per year as quality control of continuous measurements and for the analysis of other tracers and isotopes, around 25 are selected for $\Delta^{14}CO_2$ analysis to support the estimation of ffCO$_2$."

*Line 70: (ffCO2) already defined before (line 33)*

R/ We have kept only ffCO$_2$ to avoid redundancy.

*Table 1: add units to altitude (which should be elevation, actually) and sampling height*

R/ We have added the units m a.s.l. (meters above sea level) and m a.g.l. (meters above ground level) to altitude and sampling height, respectively. We have kept the term "Altitude" as it is the standard metadata terminology used within the ICOS Carbon Portal.

*Table 1 legend should say "Sampling sites included"*

R/ We have corrected the typo and changed the sentence to "Sampling stations included…" to keep consistency with Figure 1.

*Line 95: a few words about why 2018 was chosen would be nice*

R/ We have removed the reference to 2018 from this section, as the choice of year is only relevant to the data used and is already explained later in Section 2.2. Meteorological and flux data for 2018 were used to generate footprints, synthetic observations, and to perform the OSSEs. However, this year was selected primarily due to in-house data availability, rather than for any specific scientific reason.

*Line 97: space between studies and (*

R/ We have corrected all double spaces in the manuscript.

*Line 122: "6km" space between number and units*

R/ We have corrected all units typesetting.

*Line 124-125: sentence should be rephrased*

R/ We have rephrased the sentence as follows:

"The first foreground term in Eq 2b, $\mathbf{y}^f_{\Delta ff}$, represents the reduction in atmospheric $\Delta^{14}CO_2$ due to the addition of fossil $CO_2$, which is devoid of radiocarbon. This dilution effect is modeled by transporting a tracer, $\mathbf{y}^f_{\Delta ff}$, assigned a $\Delta^{14}CO_2$ value of -1000 ‰, representing fossil $CO_2$ with no radiocarbon content relative to the atmospheric standard."

*For the passage around line 130 about old carbon storage, a source would be nice*

R/ We have included as references (Graven et al., 2012; Levin & Kromer, 2004; Sweeney et al., 2007).

*Line 169: section number missing?, also check consistency between section and sec., as well as Fig. and Figure*

R/ We have added the missing reference to Sect. 3.3 and have made the use of Sect. and Fig. consistent with the Copernicus journal style.

*Line 208: OSSEs were defined already make sure you do not define abbreviations multiple times, at least in the same chapters*

R/ We have removed the repeated definition of OSSEs in this section to avoid redundancy, since the abbreviation is already introduced earlier in the manuscript.

*Line 238: this was already explained in the introduction, remove or shorten*

R/ We have removed the sentence.

*Line 244: start new sentence before "Therefore,"*

R/ We have followed the recommendation and rephrased the sentence for clarity:

"Therefore, they share the same annual budget and spatial distribution, which is defined using the location of nuclear facilities and aggregated over a 0.5° ´ 0.5° grid."

*Line 267: no space before CO2*

R/ We have corrected this instance and all other occurrences of this formatting issue throughout the manuscript.

*Line 295: this first section is a repretition from previous information, shorten or remove*

R/ While the sampling strategies are briefly introduced in the abstract, here they are presented in greater detail to explain the specific criteria used in the model implementation. We believe this clarification in the methods section is important for reproducibility and transparency, and therefore have kept the section, with minor edits for conciseness:

"We define three criteria to guide the selection of $\Delta^{14}CO_2$ flask samples in the OSSEs: (1) midday sampling at 13:00 LT every third day, (2) selection of high fossil $CO_2$ events, and (3) avoidance of periods with high nuclear emissions. These correspond to the three strategies described earlier but are detailed here with their specific operational implementation."

*Figure 2: would be nice if the months on the graphs and the tables were properly lined up*

R/ We have adjusted the x-axis formatting and repositioned the table so that the monthly tick marks and the table columns now align visually.

*Lines 364-367: not so clear how different points 3 and 4 are*

R/ We have revised the description to clarify the difference between points 3 and 4. Step 3 refers to adding observations that still meet the fossil threshold but have moderate nuclear influence, while step 4 includes observations that only meet the nuclear threshold, prioritizing those with high fossil $CO_2$ content:
"
3. If fewer than 10 such observations are available, we fill the remaining slots with observations that meet the $CO_2$ threshold and have moderate nuclear influence (1–2 ppm $C\Delta^{14}C$).

4. If this is still insufficient, we complete the sample by selecting observations that meet only the nuclear threshold, prioritizing those with the highest fossil $CO_2$ influence."

*Figure 3: address missing data for CBW in October, for b) make units more consistent with a) and c) (decimal point), also put ‰ in parenthesis*

R/ We have included in the figure the changes suggested and modified the caption as follows:

"Comparison of the available real $\Delta^{14}CO_2$ integrated samples (black) with the modeled synthetic observations (teal) at a) CBW, b) KRE and c) OPE, three of the sampling sites selected for the intensive CORSO flask campaign. The nuclear (red) and terrestrial disequilibrium (yellow) components of the synthetic observations are also shown for comparison. Gaps in panel a reflect periods of missing integrated observations. At CBW, the integration period was approximately one month during 2018, whereas it was around 14 days at the other stations. Synthetic observations were modeled to match fixed 14-day integration periods across sites."

*Line 416: WCE not defined before*

R/ We have defined WCE at an earlier mention to Western/Central Europe (WCE) and removed the redundant definitions in subsequent paragraphs.

*Line 475: I would remove the first paragraph entirely, or at least merge it with the later paragraph which discusses nuclear influence as well*

R/ We agree with the referee. We have removed the redundant paragraph from the discussion and included its main points into the opening of the discussion section. This allows us to retain the key messages regarding nuclear masking and the supporting literature, while avoiding repetition of details already presented in the results.

*Line 539: the sentence should also indicate that a frequent sampling will be more representative as well*

R/ We have included the referee's suggestion.

*Figure 8: would a zoom on Switzerland be relevant, do you think?*
*Figure 8 (legend): July 30th*

R/ We have removed Figure 8 and revised this section of the discussion to remove detailed results and instead focus on the broader interpretation of the findings. The revised paragraph highlights the limitations of integrated sampling in nuclear-influenced regions, using JFJ as a case study, and emphasizes the need for site-specific strategies.

*Line 556 and 557: right-alignment problem*

R/ We have solved this issue.

*Lines 589 and 592: VSI is defined twice*

R/ We have removed the second definition.

**RC2: 'Comment on egusphere-2024-3013', Anonymous Referee #1**

*General Comments*

*The authors present useful simulations and analysis exploring the constraints on fossil fuel emissions over Europe from atmospheric observations and inversion modelling. The written presentation in the paper could be clarified and some statements and choices in approach could be better justified. Please see specific comments below.*

R/ We thank the referee for their constructive feedback. We have carefully revised the manuscript to improve clarity throughout the text. We have also expanded our justifications for key methodological choices and clarified several statements to enhance the overall coherence of the study. Specific responses to individual comments are provided below.

*Specific Comments*

*It's a bit strange to say you are preparing for a campaign that has already being done, presumably they had to select the strategy already? Perhaps the title and some text to this effect should be rephrased.*

R/ We agree with the referee. The aim of this manuscript was to document the methodology developed during the CORSO project prior to the start of the sampling campaign and before the selection of samples for radiocarbon analysis began. However, the preparation of the manuscript took longer than expected. We have acknowledged this in the introduction and have slightly adjusted the phrasing in sentences where the campaign is mentioned to reflect that it has already taken place. We have also updated the title to: "Modeling support for an extensive $\Delta^{14}CO_2$ flask sample monitoring campaign over Europe to constrain fossil $CO_2$ emissions"

*L13 Advise not to call it fossil CO2 contamination but rather content or concentration. Contamination sounds like something you want to avoid.*

R/ We have corrected this throughout the manuscript by replacing "fossil $CO_2$ contamination" with "fossil $CO_2$ content" or "concentration," depending on the context.

*L14 Can you say here how the sampling strategy can account for nuclear 14C contamination?*

R/ We have revised the sentence in the abstract to clarify how the sampling strategy addresses nuclear $^{14}CO_2$. It now reads:

"The largest reduction in uncertainty is achieved when sample selection actively avoids periods of high nuclear $^{14}CO_2$ contamination."

*L24 Does the inverse modeling enhance the distinction between fossil fuel emissions and natural biogeochemical fluxes, or do the tracer data?*

R/ We agree that the distinction between fossil and biogenic $CO_2$ is made possible by tracer data, particularly radiocarbon for this manuscript, while inverse modeling uses this information to

constrain emission estimates. We have revised the paragraph to clarify this distinction and avoid any misinterpretation:

"These tracers help distinguish fossil fuel emissions from natural biogeochemical fluxes, and their integration into inverse models provides stronger constraints on source attribution."

*L29 Incorrect reference. Not that clear how precise D14C data will help if nuclear emissions are uncertain.*

R/ We have corrected the reference and clarified the sentence to better reflect the issue. We agree that measurement precision alone does not resolve the challenge posed by uncertain nuclear emissions. Instead, our study and others highlight the importance of strategic sample selection. We have reformulated the paragraph as follows:

"However, in Europe and other industrialized regions, the ability to isolate fossil $CO_2$ using radiocarbon is complicated by the presence of radiocarbon emissions from nuclear facilities. These emissions can artificially elevate atmospheric $\Delta^{14}CO_2$ levels, masking the depletion signal caused by fossil $CO_2$ and potentially leading to biased estimates (Turnbull et al., 2009). For example, Graven & Gruber (2011) showed that in Europe, North America, and East Asia, radiocarbon from nuclear sources can offset around 20% of the depletion caused by fossil emissions, leading to attribution biases that may exceed those caused by biospheric fluxes in some areas."

*L72 Start new paragraph here. "was" carried out in 2024?*

R/ We have corrected the verb tense and started a new paragraph at the suggested location.

*L132 Needs rephrasing. It sounds like it is saying that the biospheric carbon has been stored for decades, and that the ocean carbon comes from the "bottom of the ocean". This is not correct. Actually, most of the ocean surface is enriched relative to the atmosphere now.*

R/ We have reformulated the paragraph using more accurate terminology to avoid misleading implications about carbon storage in the biosphere and ocean:

"The carbon exchanged between the biosphere, ocean, and atmosphere has an isotopic signature that can differ from that of the current atmosphere. In the terrestrial biosphere, carbon released through heterotrophic respiration may be enriched in $^{14}C$, reflecting the elevated atmospheric radiocarbon levels that followed nuclear weapons testing in the mid-20th century (Graven et al., 2012; Levin & Kromer, 2004). This enrichment introduces a positive isotopic disequilibrium between biospheric fluxes and the present-day atmosphere. In contrast, the ocean can release $^{14}C$-depleted carbon, especially from older subsurface waters that have been isolated from atmospheric exchange for decades, allowing radioactive decay to reduce their radiocarbon content below atmospheric levels (Graven et al., 2012; Sweeney et al., 2007). These opposing disequilibrium fluxes contribute to regional and seasonal variability in atmospheric $\Delta^{14}CO_2$."

*L137 What do you mean by "depleted"? Can delete this phrase.*

R/ We have corrected the phrasing and use a more suitable reference:

"The contribution of past nuclear weapons testing is now considered negligible due to its significant decline over recent decades (Kutschera, 2022), and is therefore not included."

*L142 Better to just say the CO2 concentration and Δ14CO2 at the boundaries of the domain. Also, it would be better to name the section "Background composition from TM5" rather than "Background mixing ratios (TM5)". Have the background estimates been evaluated? Are the background estimates optimized in the inversion?*

R/ We have updated the section title to the one suggested by the referee: "Background composition from TM5." We also rephrased the opening sentence for clarity, as follows:

"The background refers to the $CO_2$ mixing ratio and $\Delta^{14}CO_2$ isotopic signature of the atmosphere at the spatial and temporal boundaries of the domain."

Regarding the background estimates: they were neither evaluated nor optimized in this study. As described in lines 210–212 of the original manuscript, we conducted perfect transport OSSEs, meaning that the same background fields, prescribed fluxes, and transport model were used to generate the synthetic observations and to perform the inversion. To clarify this point, we have moved this explanation to Section 2 (The LUMIA framework), where the term "perfect transport OSSEs" is first introduced.

*L153-162 Please explain this more clearly*

R/ We have modified the explanation of the calculation of the background component with TM5 to improve clarity.

*L174 10000 particles over 2 weeks does not seem like very many. There are 336 hours in 14 days so that is only about 30 per hour.*

R/ We thank the referee for pointing this out. The original sentence was indeed unclear. We have now clarified that the FLEXPART simulations release 10000 particles per hour, not over the entire two-week period.

*L177 The inverse modeling approach rather than The inverse modeling problem?*

R/ We have modified the section's title accordingly.

*L188 Can you say more about how the clusters are defined?*

R/ We have clarified the text and added a brief explanation of how the clusters are defined, including examples of their spatial resolution. A more detailed description of the clustering methodology, including its implementation and rationale, has already been published in (Gómez-Ortiz et al., 2025), to which we now refer in the manuscript.

*L203 Is the magnitude of the cost function minimized, or only the gradient?*

R/ We have clarified the text to indicate that the optimization procedure minimizes the cost function by iteratively reducing the magnitude of its gradient. The new paragraph reads as follows:

"The iterative procedure works by adjusting $\mathbf{x}$ to minimize the cost function $J(\mathbf{x})$, which represents the mismatch between the model and the observations weighted by their respective uncertainties. The optimal solution is achieved when the gradient, $\nabla_{\mathbf{x}} J$ approaches zero, indicating that a local minimum of the cost function has been reached. This approach ensures that the final estimate of $\mathbf{x}$ provides the best possible fit to the synthetic observational data while taking into account the uncertainties in both the prior information and the observations (Rayner et al., 2019)."

*L220 For which year is the atmospheric transport? Also 2018?*

R/ The atmospheric transport is for year 2018. We mention this in L165 of the original manuscript.

*L223 Not clear what you mean by "For the selection of the D14CO2 flask samples" here*

R/ We have updated the sentence to improve clarity:

"We use a fossil CO flux product based on the same methodology described for $\mathbf{F}_{ff}^t$. This product is later used to estimate the CO enhancement from fossil fuel combustion, used as a criterion for selecting the $\Delta^{14}CO_2$ flask samples."

*L225 Please divide this sentence into 3 sentences.*

R/ We have followed the referee's recommendation and rewritten the sentence as three separate sentences to improve clarity.

*L230 What about the impact of Fbiodis? Is Fbiodis the same in the prior and truth?*

R/ Yes, $F_{biodis}$ is the same in both the prior and the truth. As stated in the manuscript, this flux is prescribed and not optimized due to the high uncertainty associated with its estimation. This decision is supported by findings from our previous work (Gómez-Ortiz et al., 2025).

*L258 And the background is the same in the prior and truth? Does the background take into account the time difference for the air to travel within the domain to the observation site?*

R/ We have clarified in a previous reply that the same background is used in both the prior and the truth. The background is calculated using the two-step approach described by (Rödenbeck et al., 2009), as described in Sect. 2.1. of the manuscript. Since both simulations are performed with the full TM5 transport model, the time required for air masses to travel from the domain boundaries to the observation sites is accounted for in the resulting background fields.

*L266 Could you add a phrase or sentence explaining the Levin 2020 methodology?*

R/ We have added a brief explanation:

"We use a CO flux product based on the same methodology as the fossil $CO_2$ product (see Sect. 3.1) to simulate the CO mixing ratio and perform the $\Delta^{14}CO_2$ sample selection following the methodology described in (Levin et al., 2020), in which elevated CO deviations from estimated background values are used as a proxy to identify periods with enhanced fossil $CO_2$ signals."

*L267 "without exceeding" – Do you mean drawn from a normal distribution with the same standard deviation as the assumed observation uncertainty, or something else?*

R/ Indeed, this is what we do in our implementation. We have added the following explanation to improve clarity:

"As a final step, we add a random perturbation to the synthetic observations $CO_2$ and $\Delta^{14}CO_2$) by drawing values from a normal distribution with mean zero and standard deviation equal to the assumed observation uncertainty. This perturbation is added to each observation to mitigate the assumption of a perfect transport model."

*L278 What is the basis for these assumed spatial and temporal correlations?*

R/ Similar correlation values have been used in previous inversion systems applied to Europe. We have now added the following sentence to the manuscript for clarification:

"The correlation length used in this study is similar to that applied in other inversion systems in Europe (Monteil & Scholze, 2021; Thompson et al., 2020; Y. Wang et al., 2018). These correlation lengths are partly informed by the spatial structure of real emission uncertainties, especially in the fossil fuel sector, where emissions from different grid cells within a country or a sector often rely on the same statistical sources and downscaling proxies (Super et al., 2020). For biospheric fluxes, spatial correlations reflect the smooth variability of ecosystem processes and help ensure that the inversion retrieves spatial patterns that are consistent with the effective resolution of the atmospheric observation network. As shown by (Munassar et al., 2022), the limited coverage of atmospheric $CO_2$ measurements in Europe contributes more to the overall posterior uncertainty than the uncertainties in biosphere models themselves."

*L281 30% is a large difference between EDGAR and ODIAC for the European domain!*

R/ Indeed, the difference is significant and is one of the main reasons we chose to use ODIAC as the prior. The decision was based on differences in total budget, as well as in the spatial and temporal distribution of emissions between the two inventories.

*L285 This is not very clear. For the control vector you need uncertainties for each cluster, right?*

R/ We have reformulated the sentence to improve clarity. The revised text now reads:

"In a synthetic setup like the one in this study, we could in principle define uncertainties based on the known differences between the prior and true fluxes. However, to better reflect the conditions of a real inversion, where the true fluxes are unknown, we define relative uncertainties at the native resolution of the prior fluxes (e.g. 0.5°, hourly), scaling them to the magnitude of the prior flux. These uncertainties are then aggregated to the resolution of the control vector (i.e. the spatial

and temporal optimization clusters), where they define the standard deviation associated with each parameter in the inversion. This method ensures that regions and time periods with larger fluxes are assigned proportionally larger uncertainties, while still allowing the model to optimize emissions in all clusters. This approach is consistent with previous implementations of the LUMIA system (Gómez-Ortiz et al., 2025; Monteil & Scholze, 2021)."

*L299 Do you mean simulated fossil CO and CO2? Since you would not know this in real time. Otherwise, do you mean total CO and fossil CO2 calculated from CO enhancements and assumed emissions factors?*

R/ In the OSSE framework, both fossil $CO_2$ and fossil CO are simulated and used as part of the sample selection criteria. We have revised the manuscript to make this explicit. This revision clarifies the distinction between the simulated variables used in this study and the practical implementation of the method using real-time atmospheric measurements. The updated sentence now reads:

"Events of high fossil $CO_2$ are identified using the simulated mixing ratios of fossil $CO_2$ and fossil CO, the latter serving as a reliable tracer due to its co-emission during combustion and lack of biological sources. While these are simulated values in the OSSE framework, in real-world applications, total $CO_2$ and total CO measurements are used in near-real time. Fossil $CO_2$ is then inferred from observed CO enhancements relative to a background, together with known emission ratios, as described by Levin et al. (2020)."

*L309 Since the German NPPs are now all shut down since April 2023, this may not be true in 2024 and later.*

R/ We agree with the referee that the shutdowns of German reactors are an important factor when interpreting the outcome of the campaign. We have now added a caveat to the results section:

"At sites not directly influenced by nuclear emissions, such as Białystok (BIK; see Fig. 1 and Table 1), this threshold represents 87% of the synthetic observations at 13:00 local time for the year 2018. In contrast, at sites with high nuclear impact, such as Karlsruhe (KIT) in Germany, it represents 41% of the synthetic observations (see Fig. 2). This estimate is based on simulations for 2018, when nearby nuclear facilities like Philippsburg 2 (shut down at the end of 2019) were still active. However, conditions during the CORSO campaign may differ significantly due to the shutdown of all German nuclear power plants in April 2023."

And further comment on this at the Discussion:

"While the OSSE simulations are based on 2018 conditions, future real-world flask samples may still be influenced by radiocarbon emissions from decommissioned facilities. For example, Philippsburg 2, a pressurized water reactor (PWR) located near Karlsruhe, was shut down in December 2019, yet reported $^{14}CO_2$ discharges rose from 33 Bq in 2018 to 7.8 GBq in 2021, according to the EU RADD database (https://europa.eu/radd/index.dox, last access: 17 June 2025). This increase is associated with a shift in chemical speciation during decommissioning: while PWRs mainly emit $^{14}CH_4$ during operation, emissions become dominated by $^{14}CO_2$ during dismantling and waste treatment activities (Kuderer et al., 2018). Thus, despite the official

shutdown of all German NPPs in April 2023, residual emissions may continue to influence $D^{14}CO_2$ observations at stations like KIT during the 2024 campaign. At the same time, while Germany has phased out nuclear energy, several other European countries are expanding their nuclear capacity. France, the UK, Finland, and others have recently built or approved new reactors. This growing heterogeneity in nuclear policy means that radiocarbon emissions will likely remain a persistent challenge for fossil fuel source attribution using $D^{14}CO_2$, and must be accounted for in future sampling strategies and inversion frameworks."

*L315 But it is not possible to evenly distribute them throughout the year while also meeting sampling criteria in some cases, e.g. your Fig 2a (note a label missing from figure). Maybe rather than essential, you could say desirable.*

R/ We have rephrased the entire paragraph to clarify this point:

"During the CORSO sampling campaign, approximately 120 flask samples (10 per month) are selected at each station for $D^{14}CO_2$ analysis. Maintaining a consistent number of samples per station and distributing them as evenly as possible throughout the year is desirable to reduce seasonal bias. However, this distribution is not always achievable when applying strict sampling criteria, particularly in regions or periods with frequent nuclear contamination or low fossil signals. Therefore, we prioritize synthetic samples that meet the selection thresholds in each OSSE and complete the 10-per-month target with additional samples that closely match the criteria."

*L349 When neither threshold is met*

R/ We have corrected the sentence.

*L388-392 Can you show evidence for the attribution described?*

R/ We have revised the paragraph to clarify the interpretation by explicitly referencing the individual tracer contributions used in our OSSE setup. In addition, we have updated Figure 3 to include the simulated components of nuclear emissions and terrestrial isotopic disequilibrium. The paragraph now reads:

"These high values may be primarily driven by nuclear emission enrichment, as indicated by the simulated nuclear component (red line; see Fig. 3), which shows contributions of up to 7 ‰ at KRE and OPE during this period. During the growing season, when heterotrophic respiration is more active, elevated values could also be influenced by terrestrial isotopic disequilibrium, as reflected in the simulated component ranging from 1 to 4 ‰."

*L416 WCE acronym not defined*

R/ We have defined WCE at an earlier mention to Western/Central Europe (WCE) and removed the redundant definitions in subsequent paragraphs.

*Section 4.2.1 Would be useful to add a sentence at the start that here you are comparing the BASE and CORSO experiments. Can a table or chart be added to summarize the numbers given? It is difficult to absorb all the numbers in the text – Also for section 4.2.2.*

R/ We have added a clarifying sentence at the beginning of Section 4.2.1 to indicate that this section compares the BASE and CORSO experiments. Additionally, we have included a new summary table (Table 3) that presents the minimum and maximum prior bias and uncertainty values, along with their corresponding posterior reductions and associated months. This aims to support the reader in navigating the numerical content more easily. A similar table is also provided for Section 4.2.2 (Table 4).

*L427 TgC day-1*

R/ We have corrected this error.

*Figure 4 caption: Does the red shaded area represent something different than the yellow and green areas for the posteriors?*

R/ We have updated the figure to improve the clarity of the visualization. Instead of line plots, we now present a bar plot where error bars are used to represent uncertainties. All error bars represent the $\pm 1\sigma$ uncertainty derived from a Monte Carlo ensemble of 25 members. The prior uncertainty is defined independently of observational constraints, while the posterior uncertainties reflect the reduction achieved through the assimilation of synthetic observations in each inversion experiment. The figure caption has been revised to clarify this distinction.

*L434 It's a bit confusing to say base case, CORSO, when you also have a BASE case. Can just say CORSO.*

R/ We agree with the referee and have removed the term "base case" to avoid confusion with the BASE experiment. The text now simply refers to "CORSO."

*For sections 4.2.1 and 2 can you give a concluding sentence that summarizes the main finding? From the figures, it looks as if the different cases are not that different.*

R/ We appreciate the suggestion and have added concluding sentences to Sections 4.2.1 and 4.2.2 to clarify the main findings. While both BASE and CORSO experiments lead to substantial improvements over the prior, their differences are generally modest. The revised text now emphasizes that CORSO shows slightly stronger performance overall, particularly in reducing uncertainty and correcting the seasonal overestimation of emissions during late spring and early summer. Similarly, in Section 4.2.2, we highlight that all three threshold-based experiments improve upon the prior, with CORSO_ffCO2 offering the most consistent bias and uncertainty reductions, especially during the summer months.

*L452 Can you compare CORSO_ffCO2_nuc14C ffCO2 emissions uncertainty to CORSO? Is there a significant difference?*

R/ We have now modified Sections 3.6.4 and 4.2.3, methods and results, respectively, to update the analysis of the experiment evaluating the impact of radiocarbon emissions from nuclear facilities. Specifically, we now compare the CORSO_ffCO2 and CORSO_ffCO2_nuc14C experiments. Our results show that the synthetic observations selected for the CORSO_ffCO2

experiment were significantly affected by nuclear emissions. Therefore, it is more appropriate to use this experiment for comparison, rather than the less realistic CORSO_ffCO2_nuc14Cmax case. We updated the experiments to make them comparable, as we now explain in Sec. 3.6.4. In summary, for each experiment, we perform a standard Monte Carlo ensemble, consistent with the other experiments, and an additional ensemble in which nuclear emissions are perturbed and added to the observation error. We focus the comparison on the posterior uncertainties, where we found a clear difference in which the CORSO_ffCO2 has a larger uncertainty attributed to the impact of the nuclear $^{14}$C emissions.

*Figure 7: This figure is confusing – why do you get 100% uncertainty reductions in Africa, Turkey and Northern Scandinavia? Please explain.*

R/ We have revised Fig. 7 to avoid any potential misinterpretation. The updated figure now shows the absolute uncertainty attributed to the impact of nuclear emissions, calculated as the absolute difference between the CORSO_ffCO2_nuc14C and CORSO_ffCO2 ensembles and their respective counterparts that include the nuclear emission perturbation.

*L475-479 Are these statements based on your footprints and simulations showing NPP influence? Why is there a bias at JFJ in Fig A2a? Are there possible biases in background D14CO2?*

R/ We have removed this paragraph since it was redundant with the results, as pointed out by Referee #2. The key messages regarding nuclear masking and relevant literature have instead been incorporated into the opening of the discussion section to maintain clarity and focus.

Regarding the bias observed at JFJ in Fig. A2a, there might be multiple sources of bias, one being the background which was calculated using optimized global fluxes for 2010 from (Basu et al., 2020), but also overestimated prior fossil emissions. However, our intention was to show that we are capable of modeling $\Delta^{14}CO_2$ isotopic ratios at each station following the seasonality.

*Discussion repeats a lot of information from Results, particularly the paragraph starting at L519. The discussion should be shortened to max 2 pages, focused on reflections and comparisons to other literature and avoiding summarizing results.*

R/ We appreciate the referee's observation. In response, we have shortened the discussion section by removing paragraphs that repeated information from the results. The revised version now focuses more clearly on the referee's recommendation.

*L532 It would be better if you had a table to show this in the results*

R/ We have added Tables 2 and 3 to the results section.

*L535 The CORSO strategy does not require any of this.*

R/ We have clarified this in the sentence:

"Implementing the selection strategy for the CORSO_ffCO2 experiments in a real-world operational […]"

*L538 Can you say that the flask samples are helpful compared to integrated samples?*

R/ We have commented about this in relation with the impact of nuclear activity:

"These findings highlight the need to adapt sampling strategies to the spatial distribution of nuclear activity. Although integrated samples are useful for capturing long-term trends, their reliability may be compromised in regions affected by nuclear emissions. In such cases, combining integrated and flask sampling, or selectively using integrated samples, can provide a more robust approach for estimating fossil $CO_2$ estimation."

*L541-563 Another long restatement of results and summary of prior work. I think the key here is to compare to the CORSO case – is CORSO_ffCO2_nuc14C any better than CORSO sampling (every 3 days)? Also, since you don't have excellent agreement with the data and you say this is from nuclear emissions, what effect does that have on your findings? I didn't see that you presented the difference between time-varying and constant emissions.*

R/ We have removed the whole paragraph between L551-562 to avoid repetition. We have also commented about the comparability of the CORSO and CORSO_ffCO2_nuc14C in a previous comment. In the respective section, we do not analyze the agreement of the posterior values with the truth, here we rather focus on analyzing the uncertainty reduction. Regarding the two sets of nuclear emissions, we did not intend to perform a direct comparison, since both sets have the same annual budget and most likely there would not be notable differences due to the perfect transport conditions. However, we perturb the time-varying emissions in the CORSO_ffCO2_nuc14C experiments to calculate the uncertainty due to the nuclear emissions. We have clarified this in section 3.6.4.

*L573-575 But do you have evidence from your experiments? Is BASE different from CORSO for Switzerland?*

R/ We do not have direct evidence from our experiments to support this statement. In the case of Switzerland, comparing BASE and CORSO would not yield substantially different results, as the only site in the region, JFJ, uses only integrated samples in both experiments. Our intention was not to suggest that BASE outperforms CORSO, but rather to acknowledge that BASE shows similar strengths in terms of uncertainty and bias reduction during periods of low fossil emissions. To properly evaluate the impact of nuclear emissions on integrated samples, a dedicated experiment comparable to CORSO_ffCO2_nuc14C, but using the BASE setup, would be needed. For this reason, we have removed all mentions of potential impacts or benefits over Switzerland and at JFJ, in order to avoid making claims with insufficient experimental support.

*L580 Is Munassar result for Europe total or some other region's posterior emissions?*

R/ Munassar et al. (2023) focused on the same European domain as our study, doing NEE only inversions. We have revised the sentence to clarify this:

"Munassar et al. (2023) found that the use of different transport models, which help us to understand the model representation error, can result in differences of 0.51 PgC yr$^{-1}$ (61%) in the posterior NEE flux estimates over Europe."

*L586 Twice as good rather than twice better*

R/ We have removed this sentence.

*L588 Represent the afternoon conditions when the planetary boundary layer (PBL) is well developed better than what?*

R/ We have removed the original sentence to avoid ambiguity. However, we retained the point about model performance during different atmospheric regimes. The revised sentence now reads:

"Moreover, model accuracy tends to be higher during well-mixed conditions (e.g., such as in the afternoon planetary boundary layer or in the free troposphere) compared to periods with stable stratification, such as during the nocturnal boundary layer or transition phases, which are more difficult to represent."

*L584-594 Why do you need to summarize the Maier study here? Can you draw it back to your study?*

R/ We have substantially reduced the summary of Maier et al. (2022) and clarified its relevance to our study by highlighting its connection to the model representation of integrated samples.

*L595-604 The key point of this paragraph is not clear. First sentence mentions prior uncertainty but does not expand on this. The rest about the summer months seems to be unavoidable at the ICOS sites based on Figure A1. How would you actually address this other than choosing different sites that have higher ffCO2 in summer?*

R/ We have revised the paragraph to expand on the role of prior uncertainty and its influence on posterior estimates. Additionally, we clarified that the persistence of high uncertainty is not limited to the summer months but is more strongly associated with regions of low sampling coverage. The updated text also emphasizes the limitations this poses for reporting emissions at subregional and subannual scales.

*L615 Do you mean that nuclear emissions are not well-known and therefore cannot be corrected for?*

R/ We have clarified the sentence to indicate that nuclear emissions are not well characterized in time, which limits our ability to correct for them. The sentence now reads:

"Therefore, we recommend prioritizing the selection of $\Delta^{14}CO_2$ flask samples based on their potential nuclear contamination, given the limited knowledge about the temporal emission profiles of most nuclear facilities in our model domain."

*Appendix A: title of Observations is not really accurate.*

R/ We have replaced the title of Appendix A by: "Additional site-level time series for $\Delta^{14}CO_2$ synthetic and observed samples"

*Figure A2 – Could there be contributions to errors in model vs obs from errors in background? Why is JFJ so different, since it's mostly a background site?*

R/ As previously explained, there might be multiple sources of bias, most likely background which was calculated using optimized global fluxes for 2010 from (Basu et al., 2020). However, our intention was to show that we are capable of modeling $\Delta^{14}CO_2$ isotopic ratios at each station following the seasonality.